# Mechanical overstimulation causes acute injury and synapse loss followed by fast recovery in lateral-line neuromasts of larval zebrafish

Melanie Holmgren[1], Michael E Ravicz[2,3], Kenneth E Hancock[2,3], Olga Strelkova[2,3], Dorina Kallogjeri[1], Artur A Indzhykulian[2,3], Mark E Warchol[1,4], Lavinia Sheets[1,5]*

[1]Department of Otolaryngology, Washington University School of Medicine, St Louis, United States; [2]Eaton-Peabody Laboratory, Massachusetts Eye and Ear, Boston, United States; [3]Department of Otolaryngology–Head and Neck Surgery, Harvard Medical School, Boston, United States; [4]Department of Neuroscience, Washington University School of Medicine, St Louis, United States; [5]Department of Developmental Biology, Washington University School of Medicine, St. Louis, United States

**Abstract** Excess noise damages sensory hair cells, resulting in loss of synaptic connections with auditory nerves and, in some cases, hair-cell death. The cellular mechanisms underlying mechanically induced hair-cell damage and subsequent repair are not completely understood. Hair cells in neuromasts of larval zebrafish are structurally and functionally comparable to mammalian hair cells but undergo robust regeneration following ototoxic damage. We therefore developed a model for mechanically induced hair-cell damage in this highly tractable system. Free swimming larvae exposed to strong water wave stimulus for 2 hr displayed mechanical injury to neuromasts, including afferent neurite retraction, damaged hair bundles, and reduced mechanotransduction. Synapse loss was observed in apparently intact exposed neuromasts, and this loss was exacerbated by inhibiting glutamate uptake. Mechanical damage also elicited an inflammatory response and macrophage recruitment. Remarkably, neuromast hair-cell morphology and mechanotransduction recovered within hours following exposure, suggesting severely damaged neuromasts undergo repair. Our results indicate functional changes and synapse loss in mechanically damaged lateral-line neuromasts that share key features of damage observed in noise-exposed mammalian ear. Yet, unlike the mammalian ear, mechanical damage to neuromasts is rapidly reversible.

*For correspondence:
sheetsl@wustl.edu

Competing interest: The authors declare that no competing interests exist.

## Introduction

Hair cells are the sensory receptors of the inner ear and lateral-line organs that detect sound, orientation, and motion. They transduce these stimuli through deflection of stereocilia, which opens mechanically gated cation channels (*LeMasurier and Gillespie, 2005*; *Qiu and Müller, 2018*) and drives subsequent transmission of sensory information via excitatory glutamatergic synapses (*Glowatzki and Fuchs, 2002*). Excessive mechanical stimulation, such as loud noise, can damage hair cells and their synaptic connections to afferent nerves. The degree of damage depends on the intensity and duration of the stimulus, with higher levels of traumatic noise producing structural damage to sensory epithelia, including hair cells (*Cho et al., 2013*; *Nordmann et al., 2000*; *Slepecky, 1986*) and lower levels leading to various pathologies, including damage to the hair-cell mechanotransduction complex/ machinery and stereocilia (*Gao et al., 1992*; *Husbands et al., 1999*), misshapen hair cells (*Bullen*

*et al., 2019*), synaptic terminal damage, neurite retraction, and hair-cell synapse loss (*Fernandez et al., 2020*; *Henry and Mulroy, 1995*; *Kujawa and Liberman, 2009*; *Puel et al., 1998*). In addition to directly damaging hair cells, excess noise also initiates an inflammatory response (*Hirose et al., 2005*; *Kaur et al., 2019*). Such inflammation is mediated by macrophages, a class of leukocyte that responds to injury by clearing cellular debris and promoting tissue repair (*Wynn and Vannella, 2016*).

The variety of injury and range of severity suggests multiple mechanisms are involved in hair-cell organ damage associated with exposure to strong stimuli. Also unknown are the cellular processes mediating repair following such trauma. While hair cells show a partial capacity for repair of stereocilia and synaptic connections, some sub-lethal damage to hair cells is permanent. Numerous studies of the mammalian cochlea suggest that a subgroup of inner hair-cell synapses are permanently lost following noise exposure (*Cho et al., 2013*; *Hickman et al., 2018*; *Kujawa and Liberman, 2009*; *Shi et al., 2013*). Glutamate excitotoxicity is likely the pathological event that initiates noise-induced hair-cell synapse loss (*Hu et al., 2020*; *Kim et al., 2019*; *Puel et al., 1998*), but the downstream cellular mechanisms are still undefined. Further, the cellular mechanisms that promote hair-cell synapse recovery following damage-induced loss are also not understood.

Zebrafish have proven to be a valuable model system for studying the molecular basis of hair-cell injury and repair. Zebrafish sensory hair cells are structurally and functionally homologous to mammalian hair cells (*Coffin et al., 2004*; *Kindt and Sheets, 2018*; *Sebe et al., 2017*). In contrast to other vertebrate model organisms, zebrafish hair cells are optically accessible in whole larvae within the lateral-line organs. These sensory organs, called neuromasts, contain clusters of ~14 hair cells each and are distributed throughout the external surface of the fish to detect local water movements. Zebrafish can repair and regenerate damaged tissues, including lateral-line organs (*Kniss et al., 2016*; *Xiao et al., 2015*). This capacity for organ repair combined with optical accessibility allows us to study cellular and synaptic damage and repair in vivo following mechanical trauma.

In order to model physical damage to hair-cell organs in zebrafish lateral line, we developed a protocol to mechanically stimulate the lateral line of free-swimming 7-day-old larvae. Using this protocol, we were able to induce mechanical injury to lateral-line organs that resembled the trauma observed in the mammalian cochlea following acoustic overstimulation. We observed synapse loss in a subset of neuromasts that appeared morphologically intact, as well as hair-cell loss and afferent neurite retraction in a subset of neuromasts that appeared morphologically disrupted. Hair-cell mechanotransduction, as measured by uptake of the cationic dye FM1-43, was significantly reduced after mechanical injury. We also observed an inflammatory response similar to that observed in the mammalian cochlea after noise trauma. Remarkably, mechanically induced lateral-line damage appeared to rapidly recover; hair-cell number and morphology returned to normal within 4 hr following exposure, concurrent with clearance of cellular debris by macrophages. Additionally, neuromasts showed partial recovery of afferent innervation within 2 hr following exposure and completely recovered hair-cell morphology and FM1-43 uptake within 4 hr. Cumulatively, these results support that mechanically injured neuromasts show similar features of damage observed in noise exposed ears, yet rapidly repair.

## Results

### Mechanical overstimulation of zebrafish lateral-line hair cells

To mechanically damage hair cells of lateral-line organs in free-swimming 7-day-old zebrafish, we developed a stimulation protocol using an electrodynamic shaker to create a strong water wave stimulus (*Figure 1A*). The frequency used for mechanical stimulation was selected and further verified (see Method Details) based on previous studies showing 60 Hz to be within the optimal upper frequency range of mechanical sensitivity of superficial posterior lateral-line neuromasts, which respond maximally between 10 and 60 Hz, but a suboptimal frequency for hair cells of the anterior macula of the inner ear (*Levi et al., 2015*; *Trapani and Nicolson, 2010*; *Weeg et al., 2002*). Dorsal-ventral displacement of a six-well dish at 60 Hz and acceleration of 40.3 m/s$^2$ (±0.5 m/s$^2$) created water flow and disturbance of the water surface that was strong enough to trigger 'fast start' escape responses—a behavior mediated in part by zebrafish lateral-line organs to escape predation (*Figure 1B* inset) (*McHenry et al., 2009*; *Nair et al., 2015*).

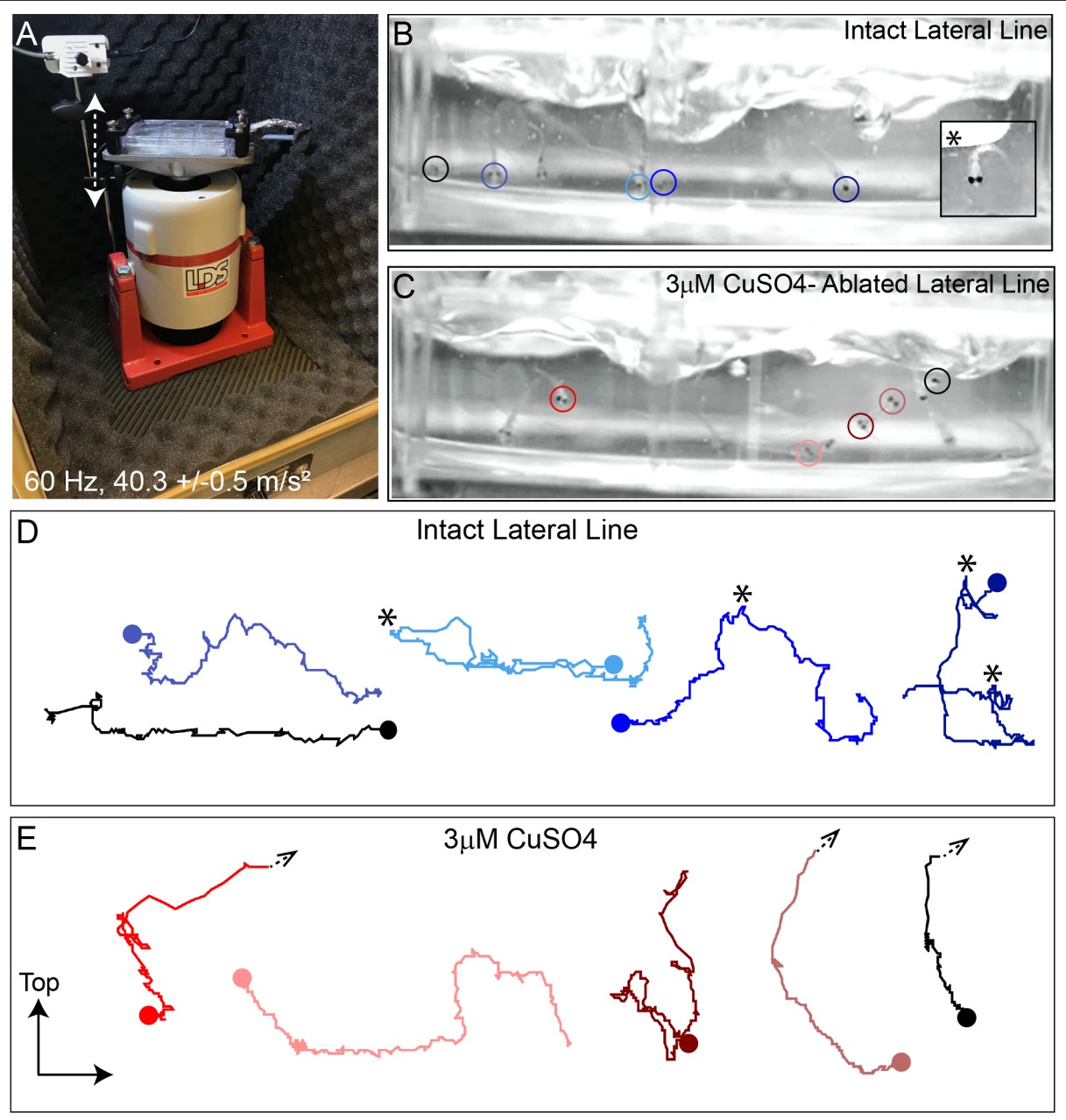

**Figure 1.** Intense water wave produced by shaker apparatus stimulates lateral-line hair cells and evokes a relevant behavior response. (**A**) The apparatus: a magnesium head expander holding a 6-well dish mounted on a vertically oriented electrodynamic shaker housed inside a sound-attenuation chamber. The stimulus consisted of a 60 Hz vertical displacement of the plate (hatched arrows) driven by an amplifier and controlled by a modified version of the Eaton-Peabody Laboratory Cochlear Function Test Suite. (**B,D**) Swimming behavior of 7-day-old larvae during exposure to the wave stimulus. Traces in (**D**) represent tracking of corresponding circled fish over 500 ms (1000 fps/ 500 frames). Asterisks indicate a 'fast escape' response (**B**; inset). (**C,E**) Swimming behavior of larvae whose lateral-line neuromasts were ablated with low-dose $CuSO_4$. Arrows in (**E**) indicate where a larva was swept into the waves and could no longer be tracked.

'Fast start' escape responses can also be activated by stimulating hair cells of the lateral line and/or the posterior macula in the ear (*Bhandiwad et al., 2013*). To verify that the observed escape responses were mediated predominantly by flow sensed by lateral-line hair cells rather than hair cells of the macula, we exposed a group of larvae to low dose (3 μM) copper sulfate ($CuSO_4$) for 1 hr to specifically ablate lateral-line hair cells, but leave hair cells of the ear intact (*Olivari et al., 2008*). Following a 2 hour recovery after $CuSO_4$ treatment, we recorded fish behavior with a high-speed camera during the stimulus (1000 fps for 10 s) and compared the responses of fish with lesioned

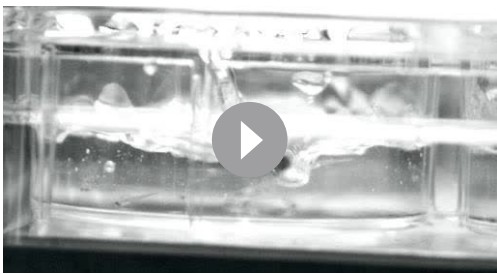

**Video 1.** Swimming behavior of control fish with intact lateral line organs. Magenta circle indicates a fish prior to a 'fast escape' response.

https://elifesciences.org/articles/69264/figures#video1

lateral-line organs with those of untreated control siblings (*Figure 1B and C*; *Video 1*). When subjected to intense water flow, we found that 'fast start' responses—defined as a c-bend of the body occurring within 15 ms followed by a counter-bend (*Burgess and Granato, 2007*; *McHenry et al., 2009*)—occurred significantly less frequently in larvae with ablated lateral line organs than in siblings with intact lateral line (*Figure 1C and E*; avg. 'fast start' responses:: 0.4 ( ± 0.1)/s in lateral-line ablated vs 1.5 ( ± 0.1)/s in control; three trials,10 s per trial; **p = 0.0043). Accordingly, some $CuSO_4$-treated fish were unable to escape the waves and were swept out of view (*Figure 1E*; arrowheads; *Video 2*). These observations indicate that the strong water wave generated by our device is stimulating lateral-line hair cells and evoking a behaviorally relevant response.

## A subset of mechanically injured neuromasts undergo physical displacement that is position dependent but does not require mechanotransduction

To induce mechanical damage to lateral line organs, larvae (10–15 per well) were exposed to an initial 20 minutes of strong water wave stimulus followed by 10 min of rest, then 2 hr of continuous stimulus (*Figure 2—figure supplement 1* A). The 10-min break in exposure was introduced early on when establishing the stimulus duration because it appeared to enhance larval survival; it was therefore maintained throughout the study for consistency. Fish were euthanized and fixed immediately after exposure, then processed for immunohistofluorescent labeling of hair cells and neurons. Unexposed sibling fish served as controls. As posterior lateral-line (pLL) neuromasts have been shown to specifically initiate escape behavior in response to strong stimuli (*Haehnel et al., 2012*), analysis of the morphology of pLL neuromasts L3, L4, and L5 was conducted for exposed and control larvae (*Figure 2A*). Initially, we divided the observed neuromast morphology into two categories: 'normal', in which hair cells were radially organized with a relatively uniform shape and size, and 'disrupted', in which the hair cells were misshapen and displaced to one side, with the apical ends of the hair cells localized anteriorly (*Figure 2B and C*; see Methods Details for measurable criteria). Position of the neuromast along the tail was also associated with vulnerability to disruption; we observed a gradient of damage in the pLL from rostral to caudal that is L5 was more susceptible to disruption than L4, which was more susceptible to disruption than L3 (*Figure 2F*; Repeated measure One-way ANOVA *p = 0.0386, **p = 0.0049, *** = 0.0004).

Additionally, we compared mechanical stress from our sustained exposure to an exposure protocol that delivered intermittent pulses of stimulus ('periodic exposure'; *Figure 2—figure supplement 1* A). We observed neuromast disruption less frequently with periodic exposures vs. sustained exposure of the same intensity (*Figure 2—figure supplement 1* B; Unpaired t-test **p = 0.0034), supporting that displacement of neuromasts is a consequence of mechanical injury. Additionally, we examined hair-cell morphology in the ears of larvae exposed to sustained stimulus and observed no apparent damage (*Figure 2—figure supplement 2*), indicating our overstimulation protocol produces mechanical damage specifically to lateral-line organs.

To determine if hair-cell activity plays a key role in the displacement of neuromasts, we exposed *lhfpl5b* mutants—fish that have intact hair-cell

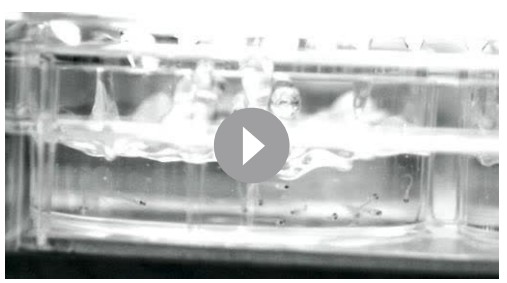

**Video 2.** Swimming behavior of larvae whose lateral-line neuromasts were ablated with $CuSO_4$. Magenta circles indicate larvae that were swept into the waves and could no longer tracked.

https://elifesciences.org/articles/69264/figures#video2

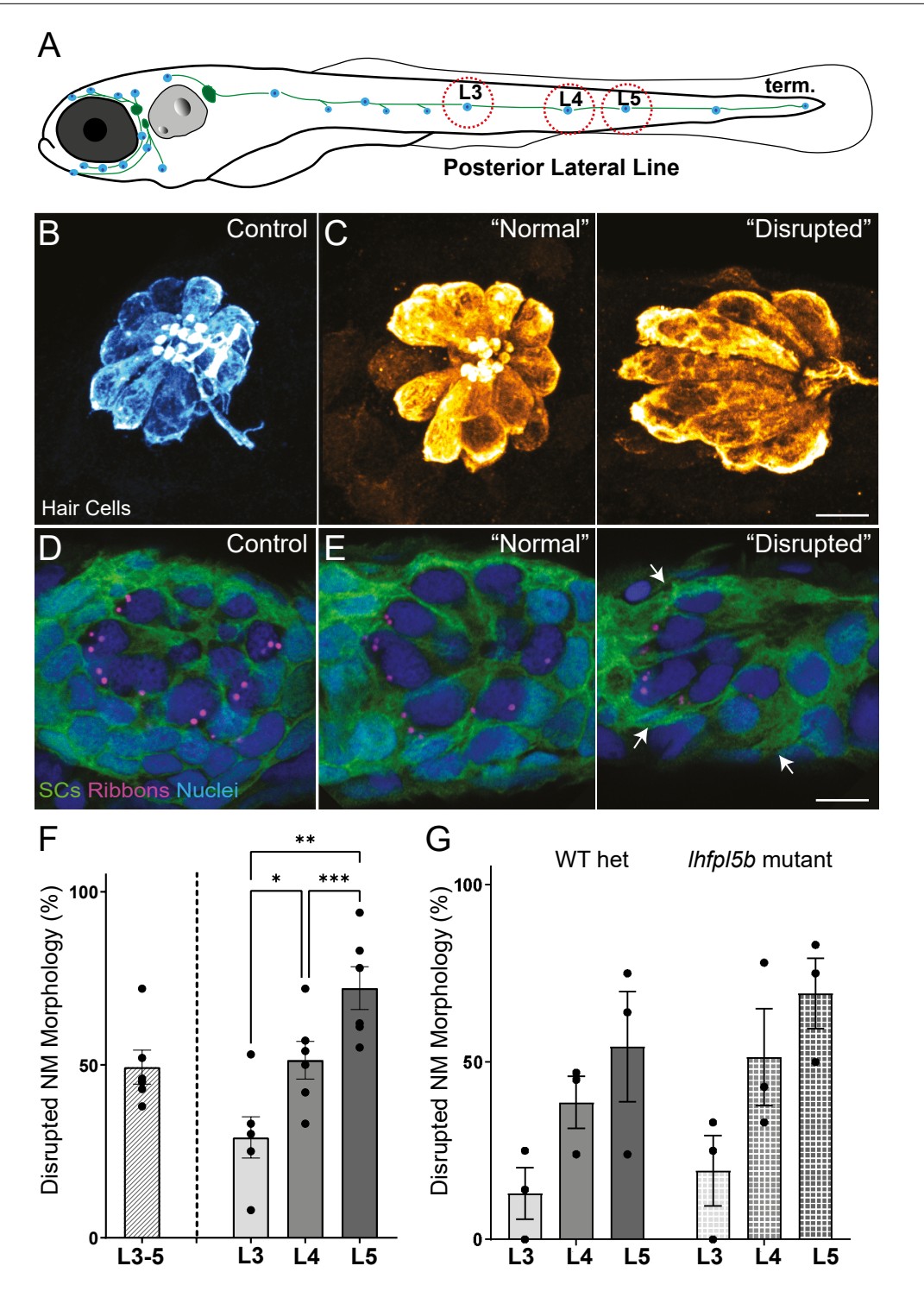

**Figure 2.** Morphological changes in pLL neuromast hair cells exposed to strong water wave stimulus. (**A**) Schematic of a larval zebrafish. Blue dots indicate neuromasts of the lateral-line organs; green lines indicate innervating afferent lateral-line nerves. pLL neuromasts L3, L4, and L5 were analyzed (dashed circles). (**B–C**) Maximum intensity dorsal top-down 2D projections of confocal images of control or stimulus-exposed neuromast hair cells (blue (**B**) or orange (**C**); Parvalbumin immunolabel). Exposed neuromast hair-cell morphology was categorized as 'normal' i.e. radial hair-cell organization indistinguishable from control or 'disrupted' i.e. asymmetric organization with the hair-cell apical ends oriented posteriorly. (**D**) Maximum intensity projections of supporting cells (SCs) expressing GFP (green), immunolabeled synaptic ribbons (magenta; Ribeye b) and all cell nuclei (blue; DAPI). Note that SCs underlying displaced hair cells also appear physically disrupted (indicated by white arrows). Scale bars: 5 µm (**F**) Average percentage

*Figure 2 continued on next page*

*Figure 2 continued*

of neuromasts with 'disrupted' morphology following mechanical stimulation. Each dot represents the percentage of disrupted neuromasts (NM) in a single experimental trial. Disrupted hair-cell morphology was place dependent, with neuromasts more frequently disrupted following sustained stimulus and when localized toward the posterior end of the tail (*p = 0.0386, **p = 0.0049, ***p = 0.0004) (**G**) Average percentage of exposed neuromasts (NM) with 'disrupted' morphology in *lhfpl5b* mutants, which lack mechanotransduction specifically in lateral-line hair cells, vs. heterozygous WT. *lhfpl5b* mutants show a similar gradient of neuromast disruption following mechanical injury as WT siblings. Error Bars = SEM.

The online version of this article includes the following source data and figure supplement(s) for figure 2:

**Source data 1.** Summary of normal and disrupted neuromast counts following sustained and periodic stimulus exposures.

**Source data 2.** ummary of normal and disrupted neuromast counts in *lhfpl5b* mutants and wildtype siblings following sustained stimulus exposure.

**Figure supplement 1.** Fish exposed to periodic stimulus have less mechanical damage to neuromasts, but still show synapse loss.

**Figure supplement 2.** Hair-cell organs of the ear appeared undamaged in larvae exposed to sustained stimulus.

function in the ear, but no mechanotransduction in hair cells of the lateral line—to sustained stimulation (*Erickson et al., 2019*). We observed comparable morphological disruption of mutant neuromasts lacking mechanotransduction (*Figure 2G*), suggesting that displacement of lateral-line hair cells is due to physical damage from the stimulus. Further, we observed the adjacent supporting cells in neuromasts with disrupted hair-cell morphology appeared similarly displaced and elongated (*Figure 2E* 'disrupted'; white arrows), indicating that mechanical injury disrupts the structural integrity of the entire neuromast organ.

## Hair-cell loss and reduced afferent innervation correspond to neuromast disruption

Moderate noise exposures can cause damage or loss of cochlear hair-cell synapses, including swelling and retraction of afferent nerve fibers, while more extended and/or severe exposures lead to hair-cell loss. To address whether mechanical damage in the lateral line produced similar morphological changes, we surveyed the number of hair cells per neuromast and the percentage of neuromast hair cells lacking afferent innervation in fish immediately following exposure to sustained stimulation (*Figure 3*). We observed a reduction in the number of hair cells per neuromast immediately following exposure (*Figure 3D and E*; **Adj p = 0.0019; N = 13 trials) as well as a significant reduction in the percentage of hair cells per neuromast innervated by afferent neurons (*Figure 3F and G*; ****Adj p < 0.0001; N = 12 trials).

As described in *Figure 2F*, we found on average~ half of L3-L5 neuromasts examined showed 'disrupted' hair-cell morphology immediately following sustained stimulus exposure. To define the associations between overall neuromast morphology and specific structural changes in mechanically injured neuromasts, we examined the numbers of hair cells and the percent of hair cells contacted by afferent fibers in exposed neuromasts parsed into 'normal' and 'disrupted' morphologies. With hair-cell number, we observed significant loss specifically in 'disrupted' neuromasts, while 'normal' neuromast hair-cell number appeared comparable to control (*Figure 3D and E*; Adj p = 0.3859 normal, ****Adj p < 0.0001 disrupted). With hair-cell afferent innervation we observed a similar trend that is, a significant number of hair cells lacked afferent innervation in stimulus exposed neuromasts with 'disrupted' morphology, but not 'normal' neuromasts (*Figure 3F and G*; Adj p = 0.7503 normal, ****Adj p < 0.0001 disrupted).

## Mechanically overstimulated neuromasts with 'normal' morphology lose a greater number of hair-cell synapses when glutamate clearance is inhibited

Hair cells contain electron-dense presynaptic specializations—called dense bodies or synaptic ribbons—apposing afferent PSDs which constitute afferent synaptic contacts (*Davies et al., 2001*; *Sheets et al., 2011*). In the larval zebrafish lateral line, afferent nerve fibers innervate multiple hair cells per neuromast forming ~3–4 synaptic contacts per hair cell. To determine whether strong water wave stimulus exposure generated lateral-line hair-cell synapse loss, we counted the number of intact synapses (ribbons juxtaposed to PSDs; *Figure 4A–C*) in control and exposed larvae. We observed significant reduction in the number of intact synapses per hair cell following sustained exposure (*Figure 4D and F*; **Adj p = 0.0078). When we compared 'normal' and 'disrupted' neuromasts

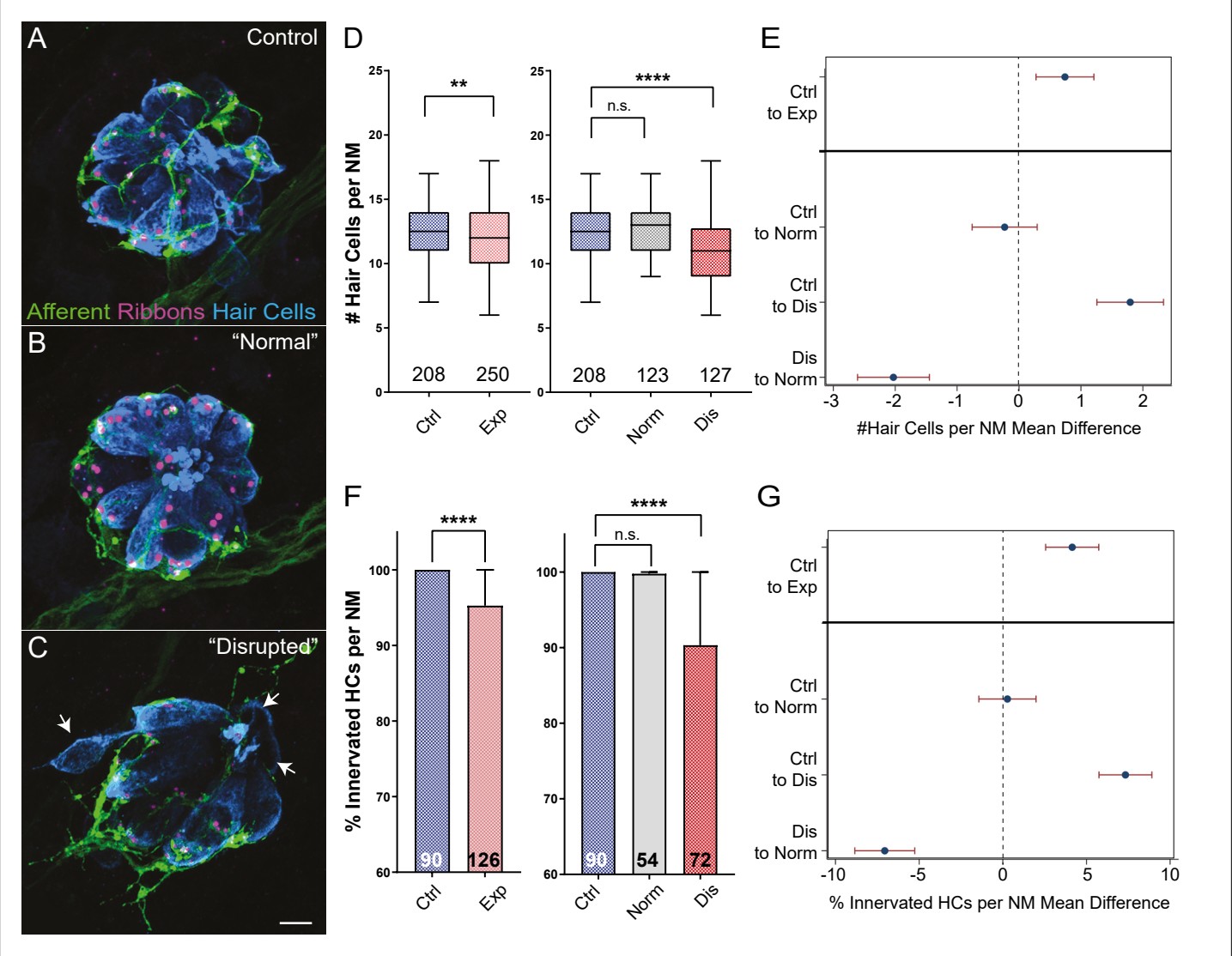

**Figure 3.** Hair-cell loss and de-innervation is specific to 'disrupted' neuromasts. (**A–C**) Representative maximum intensity projection images of control (**A**) or exposed lateral line neuromasts with 'normal' (**B**) or 'disrupted' (**C**) morphology immediately following sustained strong wave exposure (0 hr). Synaptic ribbons (magenta; Ribeye b) and hair cells (blue; Parvalbumin) were immunolabled. Afferent neurons were expressing GFP. Scale bar: 5 µm (**D**) Hair-cell number per neuromast immediately post exposure. A significant reduction in hair-cell number was observed (****Adj p = 0.0019) and was specific to 'disrupted' neuromasts (Adj p = 0.3859 normal, ****Adj p < 0.0001 disrupted). Pink box plot (Exp) represents pooled exposed neuromasts, while gray (Norm) and red (Dis) plots represent neuromasts parsed into normal and disrupted groups. Numbers beneath each plot indicate the number of neuromasts per group. Whiskers = min to max (**E**) Differences of least squares means in hair-cell number per neuromast between groups. Bars represent 95 % confidence interval (CI). (**F**) Percentage of neuromast hair cells innervated by afferent nerves. Numbers within each bar indicate the number of neuromasts per group. A significant portion of neuromast hair cells lacked afferent innervation following exposure (****Adj p < 0.0001). Hair cells lacking afferent innervation were specifically observed in disrupted neuromasts (Adj p = 0.7503 normal, ****Adj p < 0.0001 disrupted). (**G**) Differences of least squares means in % hair cells innervated per neuromast between groups. Bars represent 95% CI.

The online version of this article includes the following source data for figure 3:

**Source data 1.** Raw data and statistical analysis of hair-cell counts and innervation immediately following sustained stimulus exposure.

following exposure, we observed a loss of intact synapses per hair cell in all exposed neuromasts, with significantly fewer synapses in 'normal' exposed neuromasts relative to control (**Figure 4E–F**; **Adj p = 0.0043 normal, Adj p = 0.1207 disrupted). Additionally, significant synapse loss was observed in the neuromasts of fish exposed to the less mechanically damaging 'periodic' stimulus (**Figure 2—figure supplement 1** F').

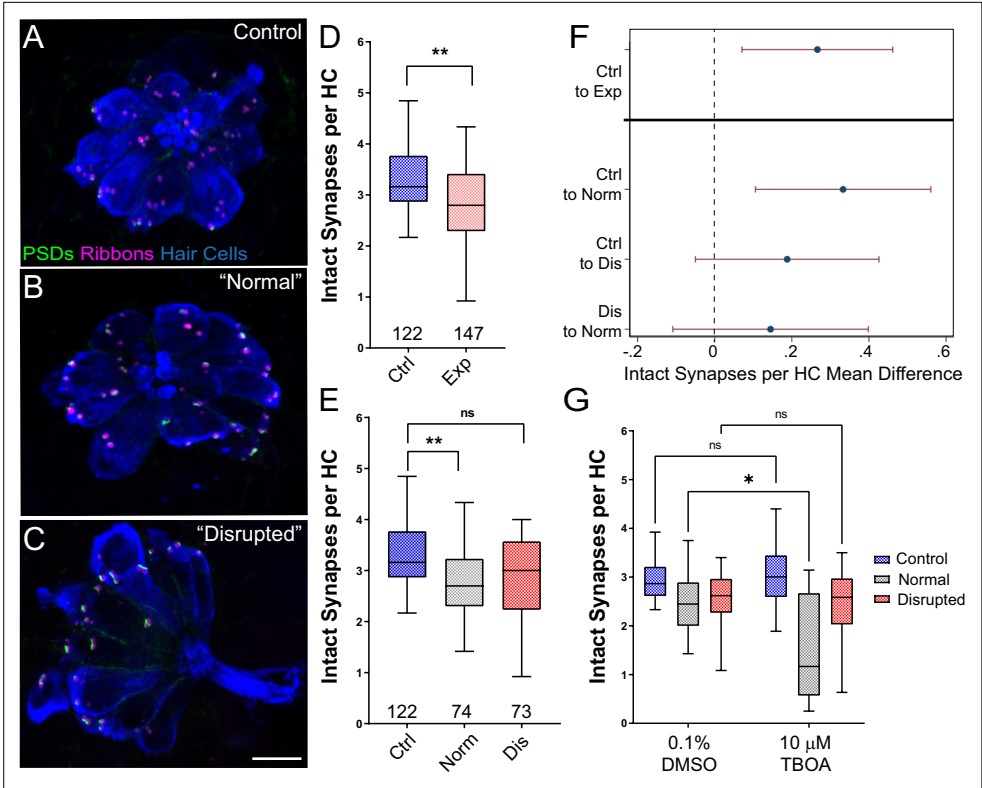

**Figure 4.** Significant hair-cell synapse loss is observed in 'normal' neuromasts following mechanical overstimulation and exacerbated by blocking glutamate uptake. (**A–C**) Representative maximum intensity projection images of unexposed (**A**), or stimulus exposed lateral-line neuromast with 'normal' (**B**) or 'disrupted' (**C**) morphology. Synaptic ribbons (magenta; Ribeye b), PSDs (green; MAGUK) and hair cells (blue; Parvalbumin) were immunolabeled. Scale bar: 5 μm (**D–E**) Intact synapses per neuromast hair cell. Pink box plot in D (Exp) represents pooled exposed neuromasts while, in E, gray (Norm) and red (Dis) plots represent neuromasts parsed into normal and disrupted groups. Whiskers = min to max. The average number of intact synapses per hair cell was significantly reduced in exposed neuromasts (D; **Adj p = 0.0078); when parsed, this reduction was significant in the 'normal' exposure group relative to control (E; **Adj p = 0.0043 normal, Adj p = 0.1207 disrupted). (**F**) Differences of least squares means in number of intact synapses per hair cell between groups. Bars represent 95% CI. (**G**) The number of intact synapses per hair cell in larvae co-treated with TBOA, to block glutamate clearance, or drug carrier alone during exposure. Synapse loss was significantly greater in 'normal' neuromasts co-exposed to TBOA compared to fish co-exposed to the drug carrier alone (Two-way ANOVA. *p < 0.0187).

The online version of this article includes the following source data for figure 4:

**Source data 1.** Raw data and statistical analysis of synapse counts immediately following sustained stimulus exposure.

Previous studies indicate that excess glutamate signaling may be a key factor driving inner hair-cell synapse loss following exposure to damaging noise (*Chen et al., 2010*; *Kim et al., 2019*). We therefore inhibited glutamate clearance from neuromast hair-cell synapses by pharmacologically blocking uptake with the glutamate transporter antagonist Threo-beta-benzyloxyaspartate (TBOA) during sustained stimulus exposure. We observed a significantly greater degree of hair-cell synapse loss in stimulated neuromasts with 'normal' morphology co-treated with TBOA than in stimulated neuromasts co-treated with drug carrier alone (*Figure 4E*; Two-way ANOVA. *p < 0.0187), suggesting glutamate excitotoxicity contributes to hair-cell synapse loss observed in mechanically overstimulated neuromasts with intact morphology.

To further characterize synapse loss in relation to afferent neurite retraction, we examined fish in which we immunolabeled synaptic ribbons, PSDs, afferent nerve fibers, and hair cells (*Figure 5A and B*). We then quantified instances where synapses, that is juxtaposed pre- and postsynaptic components associated with hair cells, were no longer adjacent to an afferent nerve terminal. As these synapses

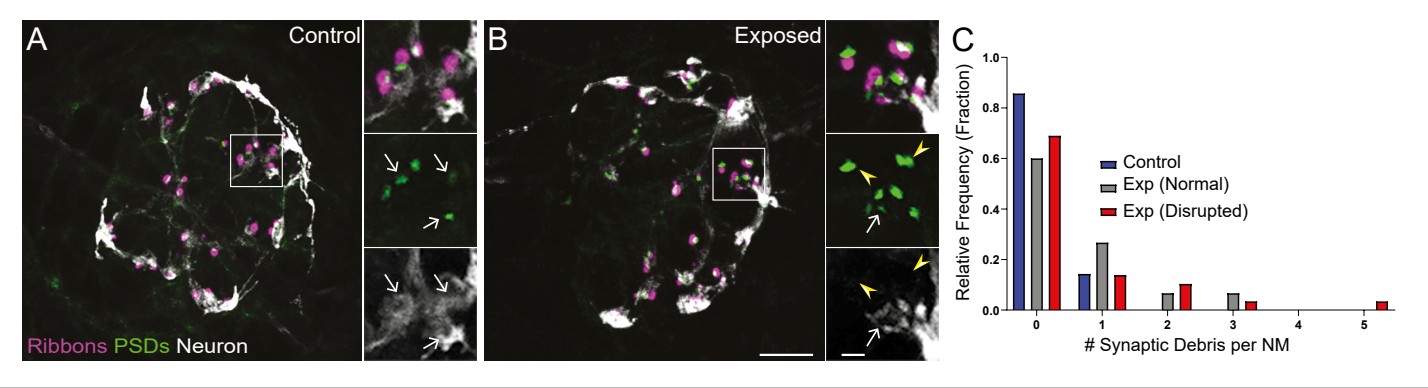

**Figure 5.** Mechanically overstimulated neuromasts showed retracted neurites and detached synaptic debris. (**A–B**) Representative images of control (**A**) and exposed (**B**) neuromasts. Synaptic ribbons (magenta; Ribeye b) and PSDs (green; MAGUK) were immunolabeled; hair cells were also immunolabeled, but not shown for clarity. Afferent neurons (white) were labeled with GFP. Insets: Arrows indicate intact synapses adjacent to afferent neurons; arrowheads (**B**) indicate synaptic debris. Scale bars: 5 µm (main panels), 1 µm (insets). (**C**) Frequency histogram of observed synaptic debris per neuromast (NM). While control neuromasts occasionally had one detached synapse, exposed neuromasts were observed that had up to five detached synapses.

appeared detached and suspended from their associated neurons, making them no longer functional, we refer to them as synaptic debris. While we rarely observed synaptic debris in unexposed neuromasts, we observed a greater relative frequency of synaptic debris in neuromasts exposed to strong water wave stimulus (**Figure 5C**; One sample Wilcoxon test, p = 0.1250 (control), *p = 0.0313(normal), **p = 0.0039(disrupted)). Taken together, we observed two distinct types of morphological damage to afferent synapses in mechanically overstimulated neuromasts: loss of synapses within neuromasts that appear intact that is exacerbated when glutamate uptake is blocked and a higher incidence of synaptic debris that appear detached from afferent neurites in all exposed neuromasts.

## Mechanically overstimulated lateral line neuromasts show signs of hair-cell injury and macrophage recruitment

The inner ears of birds and mammals possess resident populations of macrophages, and additional macrophages are recruited after acoustic trauma or ototoxic injury (**Warchol, 2019**). A similar macrophage response occurs at lateral line neuromasts of larval zebrafish after neomycin ototoxicity (**Warchol et al., 2020**). Analysis of fixed specimens, as well as time-lapse imaging of living fish (e.g. **Hirose et al., 2017**), has demonstrated that macrophages migrate into neomycin-injured neuromasts and actively phagocytose the debris of dying hair cells. To determine whether a similar inflammatory response also occurs after mechanical injury to the lateral line, we characterized macrophage behavior after sustained stimulation. These studies employed Tg(*mpeg1:yfp*) transgenic fish, which express YFP in all macrophages and microglia. Fish were fixed immediately after exposure, or allowed to recover for 2, 4, or 8 hr. Control fish consisted of siblings that received identical treatment but were not exposed to mechanical stimulation. Data were obtained from the two terminal neuromasts from the pLL of each fish (**Figure 6A**). In agreement with data shown in **Figure 3D**, we observed a modest but significant decline in hair-cell number in specimens that were examined immediately and at 2 hr after sustained exposure (**Figure 6B**; **p < 0.003). Consistent with earlier studies (**Hirose et al., 2017**), 1–2 macrophages were typically present within a 25 µm radius of each neuromast (**Figure 6C**). In uninjured (control) fish, those macrophages remained outside the sensory region of the neuromast and rarely contacted hair cells. However, at 2, 4, and 8 hr after sustained stimulus, we observed increased macrophage-hair cell contacts (**Figure 6D**; *p = 0.024), as well as the presence of immuno-labeled hair-cell debris within macrophage cytoplasm (suggestive of phagocytosis, **Figures 6A and 4** h., arrow). Macrophage-hair cell contact and phagocytosis peaked at 2 hr after exposure (**Figure 6E**; **p = 0.0013 (2 h)). Notably, the numbers of macrophages within a 25 µm radius of each neuromast remained unchanged at all time points after exposure, suggesting that the inflammatory response was mediated by local macrophages and that mechanical injury did not recruit macrophages from distant locations (**Figure 6C**). This pattern of injury-evoked macrophage behavior is qualitatively similar to

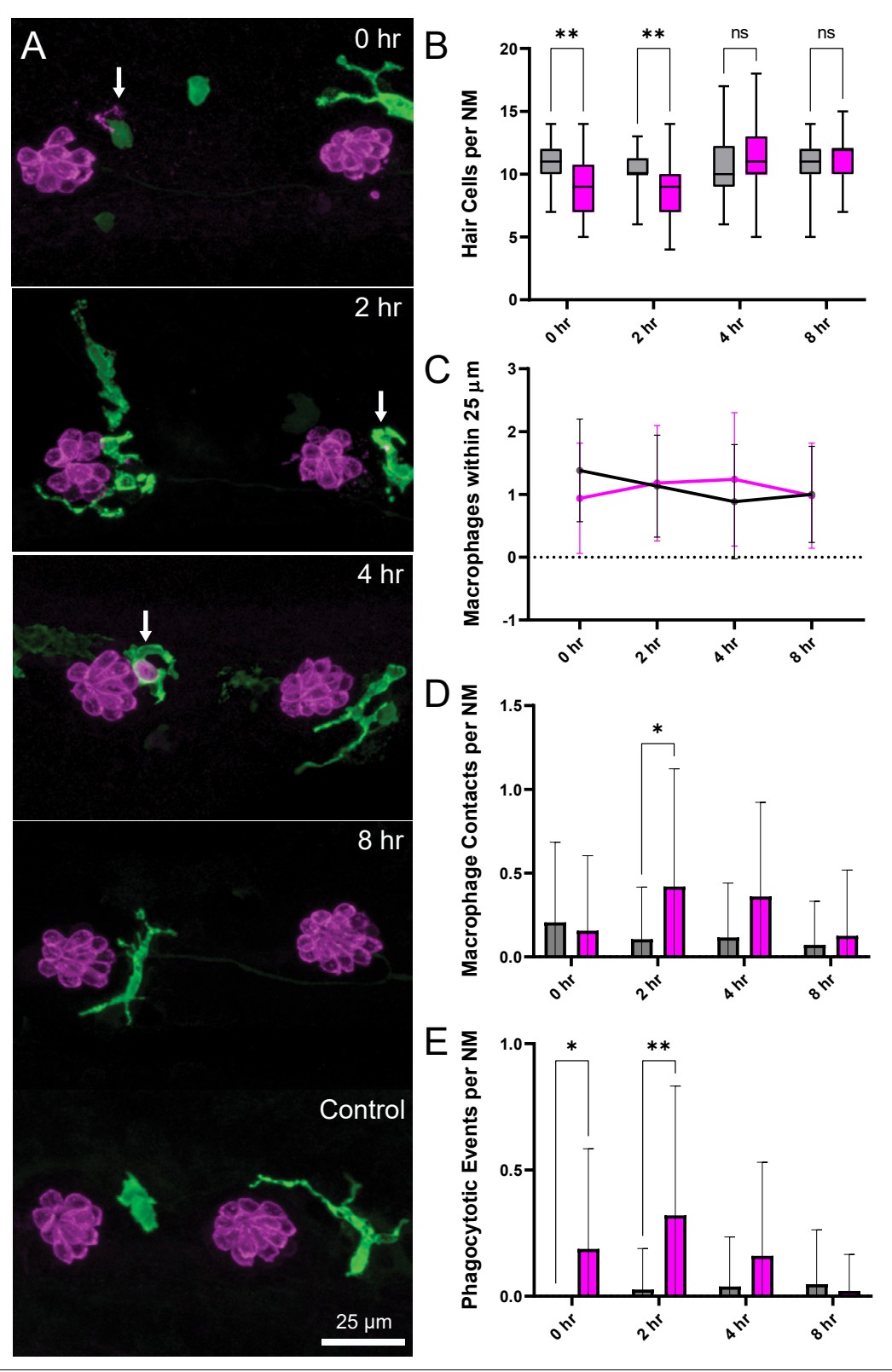

**Figure 6.** Macrophage response to mechanical overstimulation of lateral line hair cells. Experiments used Tg(*mpeg1:yfp*) fish that express YFP under regulation of the macrophage-specific *mpeg1* promoter. All images and data were collected from the two distal-most neuromasts of the posterior lateral line (*Figure 2A*; term). (**A**) Macrophages (green) responded to mechanical injury by entering neuromasts, contacting hair cells and

*Figure 6 continued on next page*

*Figure 6 continued*

internalizing Otoferlin-immunolabeled debris (arrows, magenta). Images show examples of macrophage behavior at different time points after noise trauma. (**B**) Quantification of hair-cell number in the terminal neuromasts. Hair-cell number was significantly reduced at 0–2 hr after noise exposure (Mixed-effects analysis: **p < 0.003). (**C**) Quantification of macrophages within a 25 μm radius of the neuromasts at 0–8 hr after noise injury. Most neuromasts possessed 1–2 nearby macrophages and this number was not changed by noise exposure. (**D**) Quantification of direct contacts between macrophages and hair cells. The number of macrophage-hair cell contacts was counted at each survival time after noise exposure and normalized to the total number of sampled neuromasts. Increased levels of contact were observed at 2 and 4 hr after noise (*p = 0.0243). (**E**) Quantification of phagocytosis as a function of post-noise survival time. The numbers of macrophages that had internalized otoferlin-labeled material were counted at each time point and normalized to the total number of sampled neuromasts. The percentage of macrophages that contained such debris was significantly increased at 0–2 hr after strong water wave stimulus (*p = 0.0465; **p = 0.0013). Data were obtained from 26 to 50 neuromasts/time point. Error Bars = SD.

the macrophage response observed in the mouse cochlea after acoustic trauma (*Hirose et al., 2005*; *Kaur et al., 2019*; *Kaur et al., 2015*).

## Mechanically injured neuromasts rapidly repair following exposure

To determine if damage to mechanically injured neuromasts was progressive, persistent, or reversible, we compared neuromast morphology, hair-cell number, and innervation at 0 , 2 , and 48 hr following sustained exposure to strong water wave stimulus. We observed a decrease in the percentage of neuromasts showing 'disrupted' morphology 2 hr following exposure, relative to fish fixed immediately following exposure (*Figure 7A*; 54 % disrupted (0 h) vs. 32 % disrupted (2 hr); N = 4 trials; ), suggesting that physical disruption of neuromast morphology following mechanical injury is rapidly reversible.

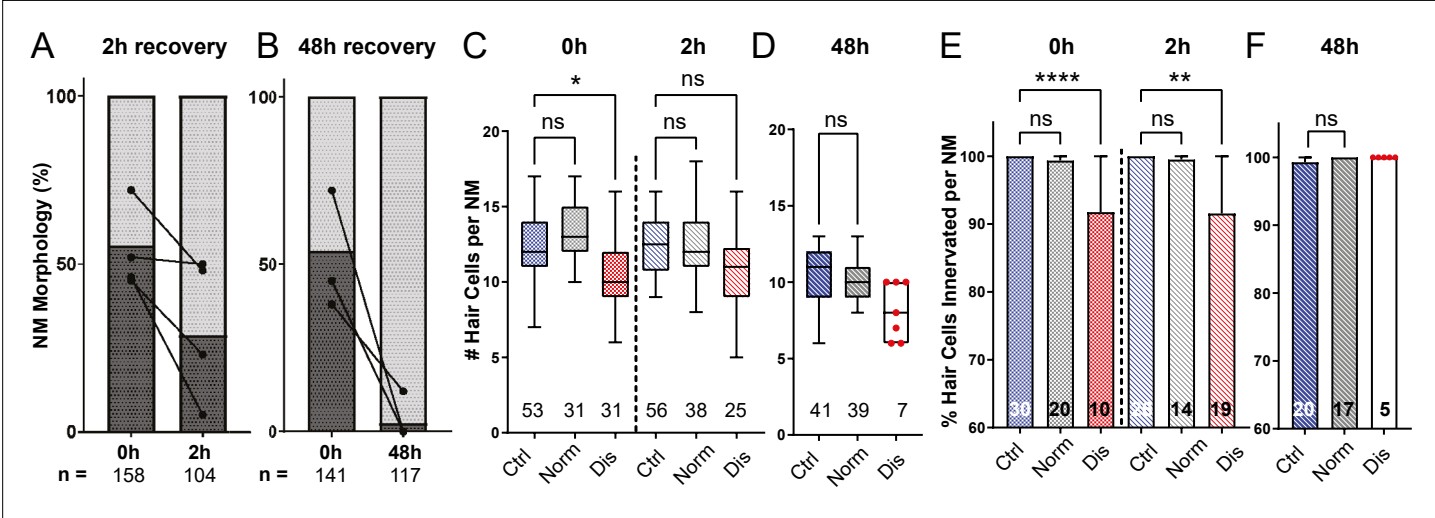

**Figure 7.** Mechanically overstimulated neuromasts recover hair-cell morphology, hair-cell number, and innervation. (**A,B**) Average percentage of exposed neuromasts with 'normal' vs. 'disrupted' morphology following exposure. Each dot represents the percentage of disrupted neuromasts (**L3–L5**) in a single experimental trial; lines connect data points from the same cohort of exposed fish following 2 hr (**A**) or 48 hr (**B**) recovery. (**C,D**) Multilevel analysis of hair-cell number per neuromast immediately (0 hr) post-exposure or after 2 or 48 hr recovery. Numbers beneath each plot indicate the number of neuromasts per group. Whiskers = min to max. Morphologically 'disrupted' neuromasts have significantly fewer hair cells at 0 hr but not 2 hr following exposure C; *Adj p = 0.0321 (0h disrupted), Adj p = 0.1875 (2h disrupted). Most exposed neuromasts were morphologically 'normal' following 48 hr recovery and had a comparable number of hair cells relative to control (D; Adj p = 0.4443). (**E,F**) The percentage of 'disrupted' neuromast hair cells lacking afferent innervation was significant following 0 hr and 2 hr recovery (E; ****Adj p < 0.0001 (0h disrupted), **Adj p = 0.0016 (2h disrupted)). All hair cells were fully innervated following 48 hr recovery, including the few neuromasts with 'disrupted' morphology (F; aligned red dots).

The online version of this article includes the following source data for figure 7:

**Source data 1.** Summary of normal and disrupted neuromast counts following sustained exposure with 0 , 2 , or 48 hr recovery.

**Source data 2.** Raw data and statistical analysis of hair-cell counts and innervation following sustained stimulus exposure with 0  and 2 hr recovery.

**Source data 3.** Raw data and statistical analysis of hair-cell counts and innervation following sustained stimulus exposure with 0  and 48 hr recovery.

Consistent with this observation, the average hair-cell number per neuromast at 2 hr post-exposure appeared to recover (*Figure 7C*; *Adj p = 0.0321 (0h disrupted), Adj p = 0.1875 (2h disrupted); N = 6 trials; ). Recovery of hair-cell number occurred within 2–4 hours (*Figure 6A and B*) and corresponded with macrophages infiltrating neuromasts and phagocytosing hair-cell debris (*Figure 6A and E*). We also observed, compared to immediately following exposure, a lesser degree of afferent fiber retraction (*Figure 7E*; ****Adj p < 0.0001 (0h disrupted), **Adj p = 0.0016 (2h disrupted); N = 6 trials) indicating partial recovery of innervation.

A recent study characterized zebrafish lateral-line hair-cell damage induced by exposure to ultrasonic waves and reported a delayed hair-cell death and synapse loss 48–72 hr following exposure (*Uribe et al., 2018*). To determine if lateral line neuromasts exposed to the strong wave stimulus generated by our apparatus underwent delayed hair-cell loss, we examined hair-cell morphology, number, and innervation 48 hr following sustained stimulus exposure. Most exposed neuromasts examined showed 'normal' HC morphology *Figure 7B*; with no significant difference in hair-cell number (*Figure 7D*; Adj p = 0.4443; N = 4 trials). Hair-cell afferent innervation after 48 hr was comparable to control fish; even the few neuromasts that remained morphologically 'disrupted' were fully innervated (*Figure 7F*).

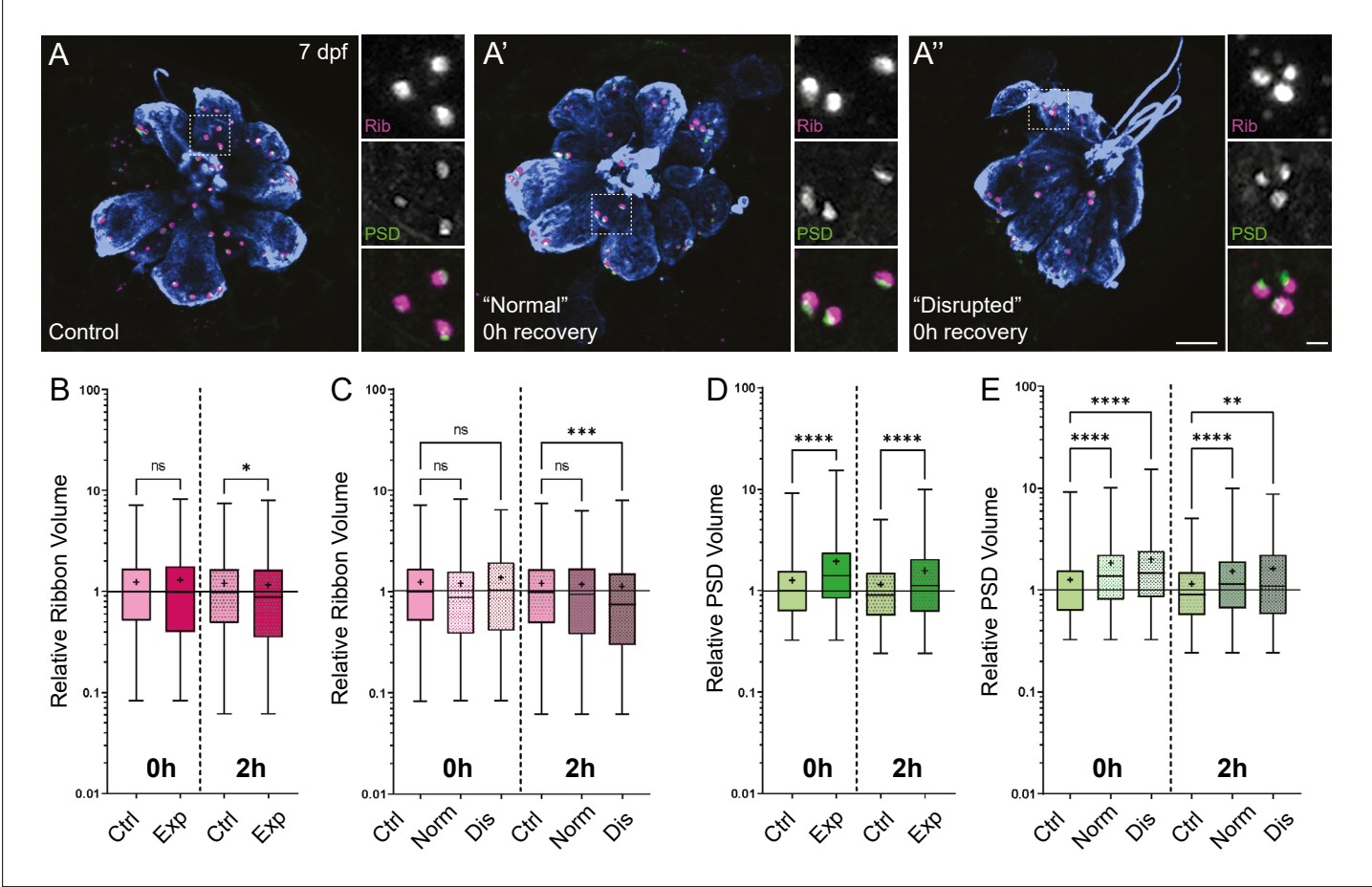

**Figure 8.** Changes in synaptic ribbon and PSD sizes following sustained mechanical overstimulation. (**A-A"**) Representative images of control (**A**) and exposed (**A', A"**) neuromasts. Synaptic ribbons (magenta; Ribeye b), PSDs (green; MAGUK), and hair cells (blue, Parvalbumin) were immunolabeled. Scale bars: 5 μm (main panels), 1 μm (insets). (**B–E**) Box and whisker plots of relative synapse volumes normalized to 0 hr control. Whiskers indicate the min. and max. values; '+' indicates the mean value, horizontal lines indicate the relative median value of the control. (**B**) Ribbon volume appeared comparable to control immediately following exposure but was reduced 2 hr after exposure (*p = 0.0195). (**C**) Significant reduction in ribbon size relative to control was specific to disrupted neuromasts (Kruskal-Wallis test: ***p = 0.0004 (2h)). (**D**) Significantly larger PSDs were observed both immediately and 2 hr following exposure (****p < 0.0001). (**E**) Enlarged PSDs were present in both 'normal' and 'disrupted' exposed neuromasts, with a greater enlargement observed 0 hr post-exposure (Kruskal-Wallis test: ****p < 0.0001 (0h); ***p = 0.0001, **p = 0.0024 (2h)).

## PSDs are enlarged in all neuromasts following mechanical overstimulation

Previous studies in mice and guinea pigs indicate moderate noise exposures modulate the size of synaptic components (*Kim et al., 2019*; *Song et al., 2016*). To determine if pre- and postsynaptic components were also affected in our model, we compared the relative volumes of neuromast hair-cell presynaptic ribbons and their corresponding PSDs in control and stimulus exposed larvae. We observed a moderate reduction in synaptic-ribbon size following exposure; ribbon volumes were significantly reduced relative to controls following 2 hr recovery (*Figure 8B*; Kruskal-Wallis test *p = 0.0195; N = 3 trials), and this reduction was specific to 'disrupted' neuromasts (*Figure 8C*). While the changes in ribbon volume we observed were modest and delayed in onset, we saw dramatic enlargement of PSDs immediately and 2 hr following exposure (*Figure 8D*; Kruskal-Wallis test ****p < 0.0001; N = 3 trials). In contrast to the observed reduction in ribbon size, relative PSD volumes were significantly enlarged in all exposed neuromasts regardless of whether neuromast morphology was 'normal' or 'disrupted' (*Figure 8E*). These data reveal enlarged PSDs as the predominant structural change in mechanically overstimulated neuromast hair-cell synapses.

## Mechanically injured neuromasts have damaged kinocilia, disrupted hair-bundle morphology, and reduced FM1-43 uptake immediately following exposure

An additional consequence of excess noise exposure in the cochlea is damage to mechanosensitive hair bundles at the apical end of hair cells and, correspondingly, disruption of mechanotransduction (*Wagner and Shin, 2019*). Larval zebrafish lateral-line hair cells each have a hair bundle consisting of a single kinocilium flanked by multiple rows of actin-rich stereocilia (*Kindt et al., 2012*). To determine if our exposure protocol damaged apical hair-cell structures, we used confocal imaging and scanning electron microscopy (SEM) to assess hair bundle morphology in both unexposed control larvae and larvae fixed immediately following sustained exposure. All neuromasts throughout the fish were evaluated, but to remain consistent with our fluorescence imaging results, we closely assessed the appearance of the caudal pLL neuromasts. We found the caudal neuromasts to be more damaged than the ones positioned more rostrally: the frequency of neuromasts with apparently disrupted appearance increased the closer its position to the tail (*Figure 9—figure supplement 1*). This is consistent with our fluorescence observations (*Figure 2F*; *Figure 9—figure supplement 2* B) in which L5 neuromasts were more likely to be disrupted than more anteriorly positioned L3.

A closer examination of neuromast morphology revealed a difference of the kinocilia length and bundling. The neuromasts of the control fish carry a bundle of long (10–15 μm), uniformly shaped kinocilia (*Figure 9A–E*). In contrast, the neuromasts of the fish fixed immediately after sustained exposure often appear to carry much shorter kinocilia (*Figure 9F–H*, yellow arrows), which lack bundling and, in some cases, pointing to different directions (*Figure 9G and H*). The apparent kinocilia length difference between control and overstimulated neuromasts suggests at least some kinocilia may undergo a catastrophic damage event at the time of stimulation, as their distal parts break off the hair cells. This is further supported by some examples of kinocilia with thicker, 'swollen' proximal shafts closer to the cuticular plate of the cell, some of which extend into a thinner distal part while others appear to lack the distal part completely (*Figure 9I–K*, yellow arrowheads). Accordingly, the average diameter of kinocilia at the level of the hair bundle (L2-5) was significantly larger than control, with a few kinocilia showing dramatically thicker widths ~ 2 x greater than the thickest control (*Figure 9L*; ***p = 0.0007). When measured ~3–5 μm above the bundle, the exposed neuromast kinocilia have a somewhat larger average diameter relative to control, but not as dramatic as observed at the base (*Figure 9M*; *p = 0.0243). Stereocilia bundles from both groups of animals carried tip links, but we were unable to systematically evaluate and quantify their abundance. However, we observed signs of damaged bundle morphology following overstimulation, as they often appeared splayed, with gaps between the rows of stereocilia (*Figure 9I–K*).

To evaluate the effect of mechanical overstimulation on hair bundle function in relation to hair bundle morphology, we assessed mechanotransduction by briefly exposing free-swimming larvae to the fixable mechanotransduction-channel permeable dye FM1-43X (*Holmgren and Sheets, 2021*; *Toro et al., 2015*). We treated control and mechanically overstimulated larvae with FM1-43X immediately and at several time points up to 4 hr following exposure, then fixed the larvae and co-labeled

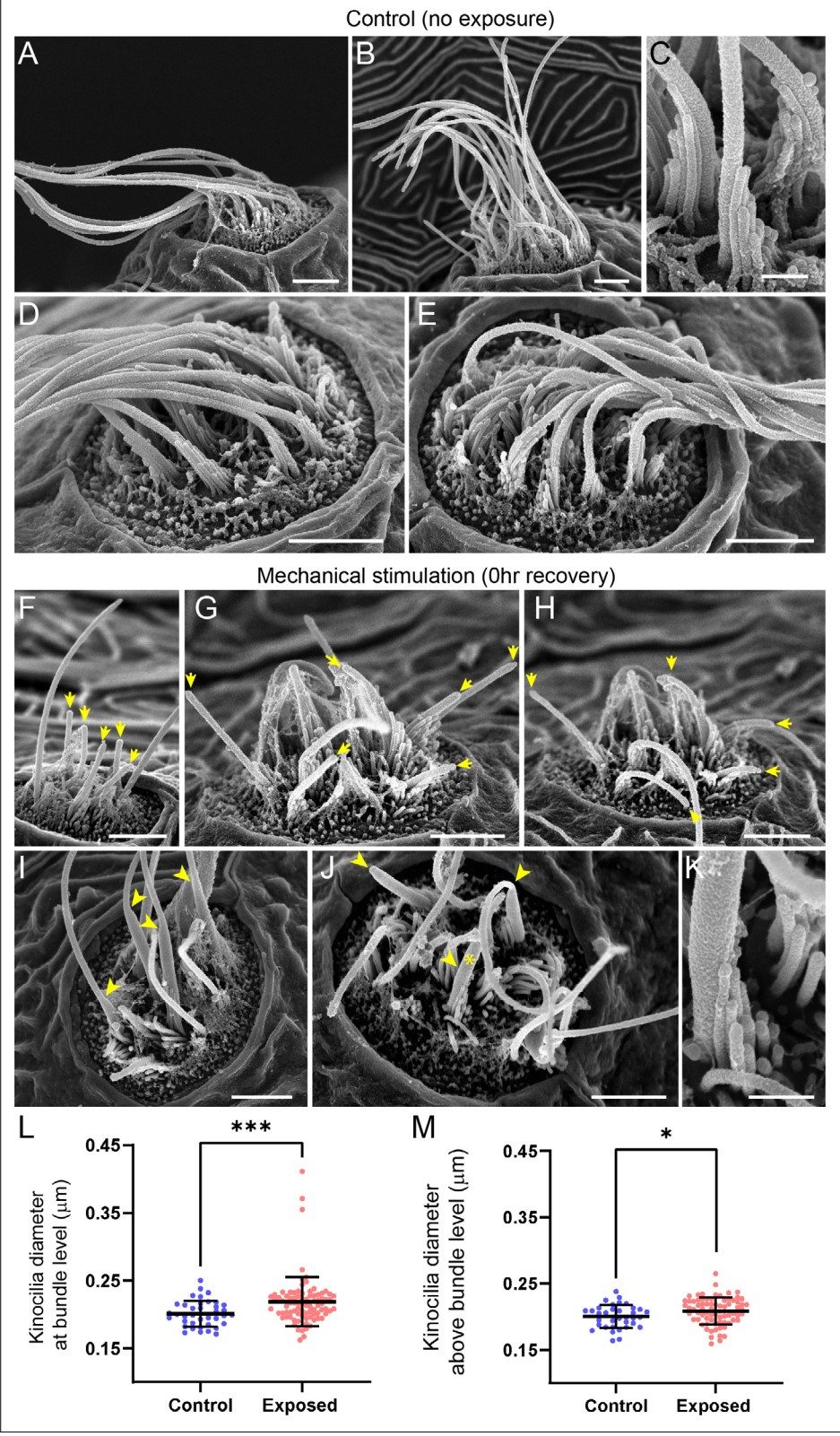

**Figure 9.** Scanning electron microscopy imaging of neuromasts following mechanical injury reveals disorganized hair-cell stereocilia bundles and damaged kinocilia. (**A–E**) Representative images of tail neuromasts of control fish larvae. Each hair cell carries a kinocilium, which is visibly thicker than its neighboring actin-filled, mechanosensitive stereocilia: see panel C featuring both structures at higher magnification (the kinocilium diameter is 220 nm, while

*Figure 9 continued on next page*

*Figure 9 continued*

stereocilia measured 90–110 nm). The kinocilia of control neuromasts are long (10–15 μm) and bundled together, while the stereocilia bundles have an apparent staircase arrangement. (**F–K**) Representative images of damaged tail neuromasts immediately following noise exposure featuring short (**F-H**, yellow arrows), disorganized (**G, H**), and swollen (**I-K**, yellow arrowheads) kinocilia, and disorganized stereocilia. (**K**) Same stereocilia bundle as in J marked with an asterisk at higher magnification to highlight the difference in the diameter of the kinocilium (360 nm) and neighboring stereocilia (85–100 nm) for noise exposed hair cells, as compared to the control hair cells in C. Scale bars: A, B, D-J – 2 μm; C, K – 500 nm. (**L–M**) Kinocilia diameter at bundle level (L; Mann Whitney test ***p = 0.0007) and 3–5 μm above bundle level (M; Welch's t test *p = 0.0243). Exposed NM data in L were not normally distributed (D'Agostino-Pearson test ****p < 0.0001). Error Bars = SD.

The online version of this article includes the following figure supplement(s) for figure 9:

**Figure supplement 1.** Scanning electron microscopy imaging of tail neuromasts following mechanical injury confirms the damage is more prominent for posterior neuromasts.

**Figure supplement 2.** Confocal images show damaged kinocilia following mechanical injury.

---

hair-cell stereocilia with fluorescently conjugated phalloidin (*Figure 10A–C*). We observed a significant reduction in the relative intensity of FM1-43 in all exposed neuromasts immediately following exposure (*Figure 10* D and F; ****p < 0.0001). While phalloidin labeling of stereocilia revealed what appeared to be tapered hair bundles in some exposed neuromasts (*Figure 10B'*; yellow arrows), average stereocilia length obtained from 3D interpolated confocal image stacks was not significantly altered (*Figure 10E*). Remarkably, FM1-43FX uptake showed recovery within 30 min and fully recovered over several hours. (*Figure 10C and D*). The degree of FM1-43FX fluorescence recovery following mechanical damage appeared to correspond with recovery of neuromast morphology; following 4 hr, nearly all neuromasts exposed to strong wave stimulus showed 'normal' morphology and relative FM1-43X fluorescence that was comparable to control (*Figures 10D, F and 4h* recovery). This observed timeline of morphological recovery coincides with macrophage recruitment and phagocytosis hair-cell debris peaking 2 hr post exposure (*Figure 6D and E*) followed by full recovery of hair-cell number between 2 and 4 hr post exposure (*Figure 6B*). Additionally, we quantified proliferating neuromast cells by treating control and mechanically overstimulated to 5-ethynyl-2-deoxyuridine (EdU) larvae for 4 hr following exposure and saw no difference in the number of EdU positive cells per neuromast in each condition (*Figure 11A–C*). Cumulatively, these data support that most damaged hair cells are repaired within mechanically injured neuromasts.

## Discussion

To model mechanical injury resulting from noise trauma in the zebrafish lateral line, we describe here a method to mechanically overstimulate neuromasts of the posterior lateral line. Using this method, we observed: (i) hair-cell synapse loss in a subset of stimulus exposed neuromasts with intact morphology, (ii) morphological displacement, hair-cell loss, and afferent deinnervation in a subset of mechanically disrupted neuromasts, (iii) an inflammatory response that peaked 2–4 hr following stimulus exposure, (iv) kinocilia and hair bundle damage, and (v) reduced FM1-43 uptake in all immediately following exposure. Remarkably, mechanically injured neuromasts rapidly recover following exposure; neuromast morphology, innervation, and mechanotransduction showed significant recovery within an hour, and most neuromasts were completely recovered within 4 hr post exposure.

### Zebrafish lateral-line as a model for sub-lethal mechanical damage and noise-induced synapse loss

Mechanical damage to the zebrafish lateral line induced by strong water wave stimulus is observable immediately following exposure, is specific to the lateral-line organ, and appears to be rapidly repaired. These observations contrast with a recently published noise damage protocol for larval zebrafish which used ultrasonic transducers (40 kHz) to generate small, localized shock waves (*Uribe et al., 2018*). They reported delayed hair-cell death and modest synapse loss that was not apparent until 48 hr following exposure, was not accompanied by decreased mechanotransduction, and was observed in the inner ear as well as the lateral-line organs. Some of the features of the damage they observed—delayed onset apoptosis and hair-cell death—may correspond to lethal damage following

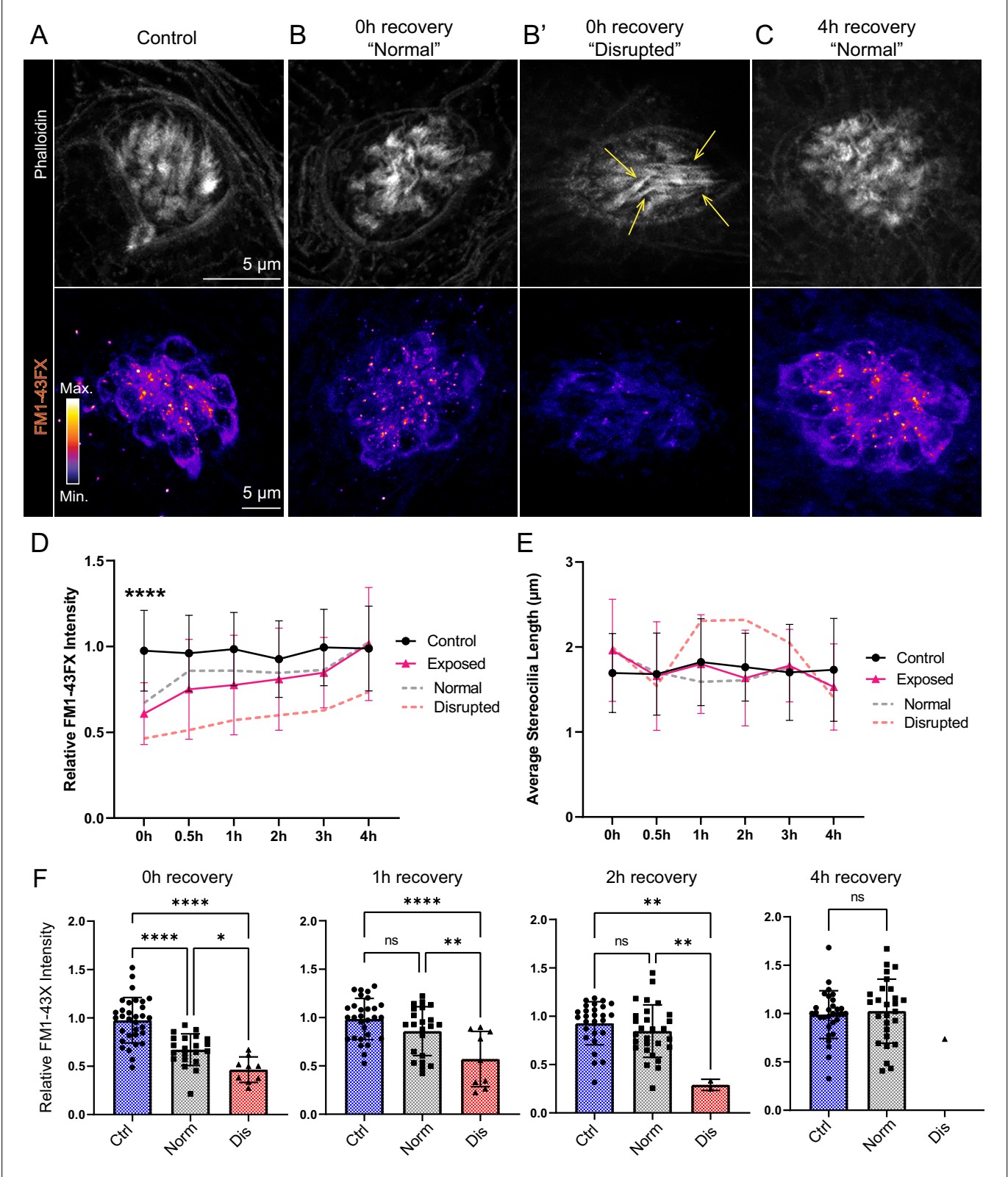

**Figure 10.** Hair-cell mechanotransduction was significantly reduced but rapidly recovered following mechanical overstimulation. (**A–C**) Representative images of hair-cell stereocilia (conjugated phalloidin, gray) and FM1-43FX fluorescence intensity of the corresponding neuromast in representative control (**A**) or mechanically overstimulated fish immediately (**B, B'**) or 4 hr (**C**) following exposure. Yellow arrows in (**B'**) indicate phalloidin labeling that appeared tapered. (**D**) Average relative FM1-43FX fluorescence intensity measurements in control and exposed neuromasts over 4 hr of recovery.

*Figure 10 continued on next page*

*Figure 10 continued*

FM1-43FX uptake was significantly reduced in exposed neuromasts immediately following mechanical overstimulation but appear to completely recover by 4 hr (Tukey's multiple comparisons test ****p < 0.0001 (0h), p = 0.0579 (1 h), p = 0.8387 (2h), p = 0.8387 (4h)). Dashed lines indicate FM1-43FX fluorescence intensity measurements in exposed neuromasts parsed into 'normal' and 'disrupted' morphologies. (**E**) Average stereocilia length of centrally localized hair bundles in control and exposed neuromasts. Dashed lines indicate measurements in exposed neuromasts parsed into 'normal' and 'disrupted' morphologies. Error Bars = SD (**F**) Relative FM1-43FX fluorescence in both 'normal' and 'disrupted' exposed neuromasts was significantly reduced immediately following exposure but recovered over time (Tukey's multiple comparisons test ****p < 0.0001, *p = 0.0328 (0h); ****p < 0.0001, **p = 0.0098 (1h); **p = 0.0025 control vs. dis, **p = 0.0089 normal vs. dis (2 h)). Each point represents an individual neuromast. Nearly all observed exposed neuromasts appeared morphologically normal following 4 hr; note only one neuromast data point in the disrupted category of the 4 hr recovery graph. Data were obtained from 26 to 32 neuromasts per condition over three trials.

blast injuries. We propose features of the damage we observe with our stimulus protocol—reduced mechanotransduction, hair-cell synapse loss, and rapid inflammatory response—may correspond to sub-lethal noise-induced damage of hair-cell organs. Because of differences in the nature of the stimuli in these two studies, it is difficult to directly compare the pathological outcomes. Mechanical overstimulation in the Uribe et al., study was induced using ultrasonic (40 kHz) actuators. Such high frequencies are far outside those that are detected by lateral line neuromasts (*Levi et al., 2015*; *Trapani and Nicolson, 2010*) supporting that cavitation in the water medium is likely causing the observed damage. In contrast, our study delivered a stimulus of high intensity 60 Hz water waves directly to the fish. This frequency is within the range of sensitivity of lateral line neuromasts of larval zebrafish and evoked a lateral-line mediated behavior (*Figure 1B and C*) suggesting that the hair cells were being directly stimulated by the water motion. The present method more closely resembles the techniques that are typically used to study noise damage in the mammalian cochlea, where high-intensity acoustic energy causes hair cell and synaptic injury in specific regions of the cochlea that are best-responsive to the frequency of the stimulus. This idea is further supported by the observation that synapse loss in hair cells exposed to strong wave stimulation is greater when glutamate uptake is blocked (*Figure 4E*), suggesting a shared mechanism of glutamate excitotoxicity between noise-exposed mammalian ears and strong water wave stimulus exposed lateral-line organs (*Kim et al., 2019*; *Sebe et al., 2017*).

## Disruption of neuromast morphology is a consequence of mechanical injury

Strong water wave exposure produced a percentage of pLL neuromasts that were morphologically 'disrupted'. Several observations support that such 'disrupted' neuromasts represent mechanical injury to neuromast organs. One is that *lhfpl5b* mutant neuromasts, which lack mechanotransduction specifically in lateral-line hair cells, were comparably vulnerable to

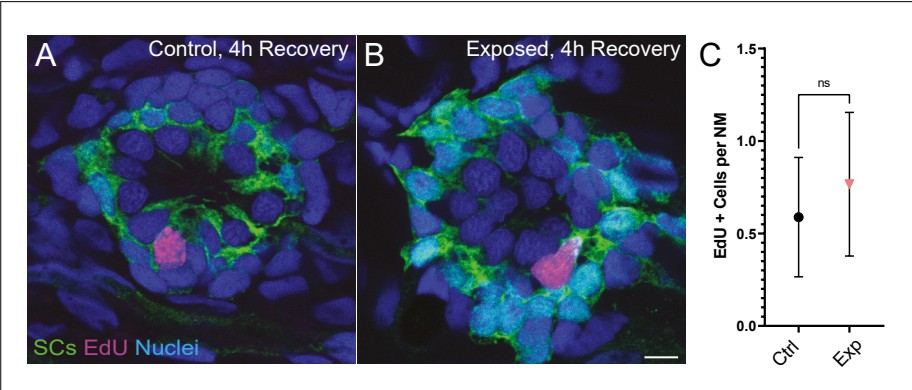

**Figure 11.** Neuromasts *show no change in cell proliferation following mechanical overstimulation.* (**A,B**) Representative cross-section images of EdU (magenta) labeling of proliferating neuromast cells. Fish were exposed to EdU for 4 hr following stimulus exposure. Supporting cells (SC) were expressing GFP. Scale bars: 5 µm (**C**) Average number of EdU + cells per neuromast were comparable in control and exposed larvae. Data were obtained from 33 to 34 neuromasts per condition over three trials (Two-way ANOVA. p = 0.4193). Bars represent 95% CI.

physical disruption as their wild-type siblings (*Figure 2G*; *Erickson et al., 2019*). This finding supports that hair-cell activity during stimulation does not underly the physical displacement of hair cells observed following strong water wave stimulus. Additionally, physical disruption of the neuromast affects the whole organ—hair cells and their adjacent supporting cells (*Figure 2D*). This observation contrasts with what is observed in mammalian ears exposed to high intensity noise, where mechanical injury to outer hair cells is localized to stereocilia disruption and gross displacement of hair cells is not found (*Wang et al., 2011*). Speculatively, displacement of hair cells in mechanically injured neuromasts may be due to loss of structural support from displaced supporting cells. As the lateral-line organs are superficially localized on the surface of the skin, some of the intense mechanical tension applied across the tail is likely coupled to neuromasts leading to the physical displacement of a subset of exposed neuromasts. Finally, we observed reduced FM1-43 uptake in 'disrupted' neuromasts (*Figure 10F*) and measurable changes to kinocilia likely reflecting mechanical damage (*Figure 9*; *Wagner and Shin, 2019*). One notable limitation to using this model is that the evaluation of subtle and functionally relevant damage to hair-cell stereocilia, such as loss of tip links and damage to the actin core, is a significant challenge in zebrafish neuromasts due to their small size. Nevertheless, it is remarkable how rapidly mechanically disrupted lateral-line neuromasts regain normal morphology and FM1-43 uptake (*Figure 10*). The superficial lateral line's direct exposure to the environment may require more robust mechanisms for repair, unlike hair-cell organs of mammals which are encased in bone.

## Hair-cell overstimulation and synapse loss

In contrast to de-innervation and modest hair-cell loss that we observed in mechanically disrupted neuromasts, we saw significant loss of hair-cell synapses in neuromasts that were exposed to strong water wave stimulus but not mechanically disrupted i.e. 'normal' (*Figure 4*). Two notable observations were made in exposed neuromasts regarding synapse loss. First, loss of synapses in 'normal' exposed neuromasts was markedly more severe when synaptic glutamate clearance was inhibited (*Figure 4G*), suggesting that synapse loss may reflect moderate hair-cell damage resulting from overstimulation of hair cells and excess glutamate accumulation. Involvement of glutamate signaling as a key mediator of noise-induced synapse loss has recently been reported in mice; loss of glutamate signaling prevents noise-induced synapse loss and pharmacologically blocking postsynaptic $Ca^{2+}$ permeable AMPA receptors protects against cochlear hair-cell synapse loss from moderate noise exposure (*Hu et al., 2020*; *Kim et al., 2019*). Second, synapse loss occurred in neuromasts that appeared to be fully innervated (*Figures 3B and 4* B). This observation was initially surprising given that pharmacologically activating evolutionarily conserved $Ca^{2+}$ permeable AMPARs has been shown to drive afferent terminal retraction in the zebrafish lateral line (*Sebe et al., 2017*). We propose that subtle damage to afferents in 'normal' exposed neuromasts may accompany synapse loss but not be apparent as loss of innervation, as single afferent processes innervate multiple hair cells of the same polarity within an individual neuromast (*Dow et al., 2018*; *Faucherre et al., 2009*). This idea is further supported by our observation that the relative frequency of synaptic debris (i.e. synaptic components that appear detached from afferent neurites) was higher in exposed neuromasts with 'normal' morphology relative to control (*Figure 5C*). We speculate that synaptic debris observed in exposed neuromasts with 'normal' morphology may be the result of excitotoxic damage at synaptic terminals (*Sebe et al., 2017*) while synaptic debris observed in 'disrupted' neuromasts may reflect mechanical disruption of supporting cells and retraction of afferent innervation from a subset of hair cells (*Figure 3C and F*).

In addition, glutamate signaling may not be the only driver of synapse loss resulting from excess stimulation. Studies in zebrafish and mice support that mitochondrial stress combined with excessive synaptic activity may also contribute to hair-cell synapse loss (*Wang et al., 2018*; *Wong et al., 2019*). Future work using this model to examine the effect of excess mechanical stimulation on mutants with reduced glutamate release or impaired glutamate clearance (to elevate or reduce glutamate in the synaptic cleft, respectively) combined with modified mitochondrial function may define the relative roles of glutamate excitotoxicity and hair-cell mitochondrial stress to synaptic loss.

## Role of inflammation following mechanical injury to lateral-line organs

Our results indicate that mechanical injury to neuromasts evokes an inflammatory response. Prior studies of larval zebrafish have shown that macrophages reside near the borders of uninjured neuromasts and migrate into neuromasts after ototoxic injury (*Hirose et al., 2017*). We found that macrophages migrate into neuromasts within ~2 hr of mechanical injury, where they contact hair cells and, in some cases, engulf hair-cell debris. Although this macrophage response is similar to that which occurs after ototoxic injury to neuromasts (*Carrillo et al., 2016*; *Hirose et al., 2017*; *Warchol et al., 2020*), the extent of hair-cell loss after mechanical overstimulation is much less than the injury that occurs after ototoxicity. We observed macrophage entry in 30–40% of exposed neuromasts, despite modest hair-cell loss (*Figure 6B and D*). It is possible that the morphological changes characteristic of mechanically injured neuromasts are accompanied by the release of macrophage chemoattractants. In addition, studies of noise exposure to the mammalian cochlea indicate that high levels of synaptic activity (without accompanying hair-cell loss) can evoke macrophage migration to the synaptic region (*Kaur et al., 2019*). In either case, the signals responsible for such recruitment remain to be identified. The observation that macrophages had internalized immunolabeled hair-cell material further suggests that recruited macrophages engage in the phagocytosis of hair-cell debris, but it is not clear whether macrophages remove entire hair cells or target specific regions of cellular injury (e.g. synaptic debris; *Figure 5*). In any case, our data indicate that the macrophage response to mechanical injury of zebrafish lateral-line neurons is similar to that which occurs after noise injury to the mammalian cochlea (*Warchol, 2019*) and suggests that zebrafish may be an advantageous model system in which to identify the signals that recruit macrophages to sites of excitotoxic injury.

## Hair-cell synapse morphology following mechanical overstimulation

Immediately following mechanical overstimulation, the most pronounced morphological change we observed in hair-cell synapses was significantly enlarged PSDs (*Figure 8D and E*). Speculatively, PSD enlargement may be a consequence of reduced glutamate release from hair cells following sustained intense stimulation. Mice and zebrafish fish lacking hair-cell glutamatergic transmission have enlarged postsynaptic structures (*Kim et al., 2019*; *Sheets et al., 2012*), indicating that glutamate may regulate postsynaptic size. While our data do not directly support this idea, we speculate that reduced glutamatergic transmission in mechanically overstimulated neuromasts may be a consequence of transiently impaired mechanotransduction (*Figure 10B and D*; *Zhang et al., 2018*). Alternatively, cholinergic efferent feedback, which has been shown to hyperpolarize lateral-line hair cells, may reduce hair-cell excitability during sustained strong wave exposure to protect against excess glutamate release, excitotoxic damage, and synapse loss (*Carpaneto Freixas et al., 2021*). Interestingly, presynaptic ribbons were not similarly enlarged, but instead showed a modest reduction in size following mechanical injury. Functional imaging of zebrafish lateral line has shown that a subset of hair cells in each neuromast are synaptically silent, and these silent hair cells can become active following damage (*Zhang et al., 2018*). As ribbon size has been observed to correspond with synaptic activity (*Merchan-Perez and Liberman, 1996*; *Sheets et al., 2012*), reduction in ribbon size may reflect recruitment of synaptically silent hair cells following mechanically induced damage. Future functional studies are needed to determine if mechanical overstimulation recruits more active hair-cell synapses, and to verify whether glutamate release from active synapses is reduced following mechanical overstimulation.

## Lateral-line neuromasts fully recover following mechanical damage

Previous studies indicate mammalian cochlear hair cells have some capacity for repair following sublethal mechanical damage, including tip-link repair and regeneration of a subset of ribbons synapses (*Indzhykulian et al., 2013*; *Jia et al., 2009*; *Kim et al., 2019*). But such ability is limited, and our understanding of hair-cell repair mechanisms is incomplete. By contrast, we observe complete recovery of neuromast morphology and innvervation following mechanical trauma to the zebrafish lateral line. The cellular mechanisms responsible for such repair are not fully defined, but may involve regulation of neurite growth, glutamate signaling, inflammation, and/or neurotrophic factors (*Kaur et al., 2019*; *Kim et al., 2019*; *Wan et al., 2014*). Further study using this zebrafish model of mechanical

overstimulation may provide insights that will assist in the development of methods for promoting complete hair-cell repair following damaging stimuli to the mammalian cochlea.

We also found that mechanical trauma resulted in a small degree of hair-cell loss in disrupted neuromasts (*Figure 3D*), but that hair-cell numbers had recovered after 2 hr (*Figure 6B*). The mechanism that mediates this recovery is not clear, but it is notable that a low level of hair-cell production normally occurs in lateral line neuromasts of larval zebrafish, as part of a process of ongoing turnover (*Cruz et al., 2015*; *Williams and Holder, 2000*). In the present study, we observed a small amount of proliferation in neuromasts of both mechanically-damaged and control fish. Since mechanical damage did not increase the level of cell proliferation in neuromasts relative to control (*Figure 11*), we believe this observed cell division is likely associated with the turnover process. Hair-cell regeneration in the vertebrate inner ear can also occur via direct phenotypic conversion of supporting cells into a replacement hair cells (*Warchol, 2011*). While it is conceivable that transdifferentiation of supporting cells could occur within 2 hr of mechanical injury, such transdifferentiation has not been previously demonstrated in zebrafish lateral line neuromasts (e.g. *Thomas et al., 2015*). Overall, these observations indicate that mechanical trauma does not increase the rate of hair-cell production within neuromasts, further supporting that neuromast recovery is largely due to hair-cell repair.

In summary, our data show that exposure of zebrafish lateral-line organs to strong water wave results in mechanical injury and loss of afferent synapses, but that these injuries rapidly recover. Our next steps will be to define the time course for synaptic recovery and to determine how lateral-line mediated behavior is affected by mechanically induced damage. Sub-lethal overstimulation of hair cells in the zebrafish lateral line provides a useful model for defining mechanisms of damage and inflammation and for identifying pathways that promote hair-cell repair following mechanically-induced injury.

# Materials and methods

**Key resources table**

| Reagent type (species) or resource | Designation | Source or reference | Identifiers | Additional information |
|---|---|---|---|---|
| Strain, strain background (*Danio rerio*) | AB | ZIRC | RRID: ZL1 ZFIN ID: ZDB-GENO-960809–7 | |
| Strain, strain background (*Danio rerio*) | Tübingen | ZIRC | RRID: ZIRC_ZL57 ZFIN ID: ZDBGENO-990623–3 | |
| Genetic reagent (*Danio rerio*) | lhfpl5b$^{vo35/vo35}$ | *Erickson et al., 2019* | RRID:ZIRC_ZL13656.05 ZFIN ID: ZDB-GENO-200824–4 | |
| Genetic reagent (*Danio rerio*) | TgBAC(neurod1:EGFP) | *Obholzer et al., 2008* | ZFIN ID: ZDB-ALT-080701–1 | |
| Genetic reagent (*Danio rerio*) | Tg(myo6b:actb1-EGFP) | *Kindt et al., 2012* | ZFIN ID: ZDB-TGCONSTRCT-120926–1 | |
| Genetic reagent (*Danio rerio*) | Tg(mpeg1:YFP) | *Roca and Ramakrishnan, 2013* | ZFIN ID: ZDB-ALT-130130–3 | |
| Sequence-based reagent | lhfpl5b_ F | *Erickson et al., 2019* | PCR primer | GCGTCATGTGGGCAGTTTTC; Made by IDT |
| Sequence-based reagent | lhfpl5b_R | *Erickson et al., 2019* | PCR primer | TAGACACTAGCGGCGTTGC; Made by IDT |
| Antibody | (Ribbon label: Mouse monoclonal anti-Ribeye b IgG2a) | *Sheets et al., 2011* | N/A | (1:10,000) |
| Antibody | (Ribbon label: Mouse monoclonal anti-panCtBP IgG2a) | Santa Cruz | Cat. No. sc-55502 | (1:1000) |

*Continued on next page*

*Continued*

| Reagent type (species) or resource | Designation | Source or reference | Identifiers | Additional information |
|---|---|---|---|---|
| Antibody | (PSD label: Mouse monoclonal anti-panMAGUK IgG1) | NeuroMab | K28/86, #75–029 | (1:500) |
| Antibody | (Chicken polyclonal anti-GFP) | Aves Labs | Cat. No. GFP-1020 | (1:500) |
| Antibody | (Hair cell label: Rabbit polyclonal anti-Parvalbumin) | Thermo Fisher | Cat. No. PA1-933 | (1:500) |
| Antibody | (Hair cell label: Rabbit polyclonal anti-Parvalbumin) | Abcam | Cat. No. ab11427 | (1:2000) |
| Antibody | (Hair cell label: Mouse anti-Otoferlin IgG2a) | Developmental Studies Hybridoma Bank | HCS-1 | (1:500) |
| Antibody | (Goat anti-Rabbit IgG Secondary Antibody, Pacific Blue) | Thermo Fisher | Cat. No. P-10994 | (1:400) |
| Antibody | (Goat anti- Mouse IgG1 Antibody, Alexa Fluor 488) | Thermo Fisher | Cat. No. A-21121 | (1:1000) |
| Antibody | (Goat anti-Chicken IgY Antibody, Alexa Fluor 488) | Thermo Fisher | Cat. No. A-11039 | (1:1000) |
| Antibody | (Goat anti-Rabbit IgG Antibody, Dylight 549) | Vector Laboratories | Cat. No. DI-1549–1.5 | (1:1000) |
| Antibody | (Goat anti-Rabbit IgG Antibody, Alexa Fluor 555) | Thermo Fisher | Cat. No. A27039 | (1:1000) |
| Antibody | (Goat anti- Mouse IgG2a Antibody, Alexa Fluor 647) | Thermo Fisher | Cat. No. A-21241 | (1:1000) |
| Peptide, recombinant protein | MluCI | New England Biolabs | Cat. No. R0538 | |
| Chemical compound, drug | DL-TBOA | Tocris | Cat. No.1223 | |
| Chemical compound, drug | Copper(II) sulfate ($CuSO_4$) | Millipore Sigma | Cat. No. 451,657 | |
| Chemical compound, drug | 2.5 % Glutaraldehyde in 0.1 M Sodium Cacodylate Buffer, pH 7.4: SEM | Electron Microscopy Sciences | Cat. No. 15,960 | |
| Chemical compound, drug | Paraformaldehyde; IHC | Millipore Sigma | Cat. No. 158,127 | |
| Commercial assay or kit | Click-iT EdU Cell Proliferation Kit for Imaging, Alexa Fluor 555 dye | Thermo Fisher | Cat. No. C10338 | |
| Software, algorithm | FIJI is just ImageJ | NIH | https://imagej.net/software/fiji/ | |
| Software, algorithm | Volocity | Quorum Technologies | https://quorumtechnologies.com/index.php/component/content/category/31-volocity-software | |
| Software, algorithm | Prism (v9) | Graphpad Software | https://www.graphpad.com/ | |
| Software, algorithm | Adobe Illustrator | Adobe | https://www.adobe.com/ | |
| Other | FM1-43X; fixable analog of FM 1–43 | Thermo Fisher | Cat. No. F35355 | 3 µM for 20 seconds |
| Other | DAPI nuclear stain | Thermo Fisher | Cat. No. Cat. No. F35355 | 5 mg/ml stock; diluted (1:2000) |

## Zebrafish

All zebrafish experiments and procedures were performed in accordance with the Washington University Institutional Animal Use and Care Committee. Adult zebrafish were raised under standard conditions at 27–29°C in the Washington University Zebrafish Facility. Embryos were raised in incubators at 29 °C in E3 media 5 mM NaCl, 0.17 mM KCl, 0.33 mM CaCl₂, 0.33 mM MgCl₂ (Nüsslein-Volhard & *Nüsslein-Volhard and Dahm, 2002*) with a 14 hr:10 hr light:dark cycle. After four dpf, larvae were raised in 100–200 ml E3 media in 250 ml plastic beakers and fed rotifers daily. Sex of the animal was not considered in our studies because sex cannot be predicted or determined in larval zebrafish.

The transgenic lines *TgBAC(neurod1:EGFP)* (*Obholzer et al., 2008*), *Tg(tnks1bp1:EGFP)* (*Behra et al., 2012*), *Tg(–6myo6b:βactin-EGFP)* (*Kindt et al., 2012*), and *Tg(mpeg1:YFP)* (*Roca and Ramakrishnan, 2013*) were used in this study. Fluorescent larvae were identified at three dpf without anesthesia in E3 media. The mutant line *lhfpl5bᵛᵒ³⁵/ᵛᵒ³⁵* was also used (*Erickson et al., 2019*).

## Genotyping

To genotype *lhfpl5bᵛᵒ³⁵/ᵛᵒ³⁵* larvae and siblings after mechanical stimulation and immunohistochemical labeling, ~ 1 mm tail tissue was excised, and genomic DNA was extracted by incubation in a lysis buffer (10 mM Tris pH 8.0, 50 mM KCl, 0.3% NP-40, 0.3 % Tween-20). A genomic region of *lhfpl5b* was amplified by PCR using forward primer GCGTCATGTGGGCAGTTTTC and reverse primer TAGACACTAGCGGCGTTGC. The *lhfpl5bᵛᵒ³⁵* mutation disrupts a MluCI restriction site (AATT), so PCR products were digested with MluCI, and homo- and heterozygotes were resolved by differences in band size on a 1–1.5% agarose gel.

## Experimental apparatus

Multi-well plates containing larvae were clamped to a custom magnesium head expander (Vibration & Shock Technologies, Woburn, MA) on a vertically oriented Brüel + Kjær LDS Vibrator, V408 (Brüel and Kjær, Naerum, Denmark). An additional metal plate was fitted to the bottom of the multi-well dish to fill a small gap between the bottoms of the wells and the head expander to eliminate flexing of the well plate relative to the head expander. Vibrometry of the well bottoms and the head expander with a laser-Dopper vibrometer (OFV-2600 and OFV-501, Polytec, Irvine, CA) confirmed that the well plate and head expander motion were equal at stimulus frequencies. This experimental apparatus was housed in a custom sound-attenuation chamber. An Optiplex 3,020 Mini Tower (Dell) with a NI PCIe-6321, X Series Multifunction DAQ (National Instruments) running a custom stimulus generation program (modified version of Cochlear Function Test Suite) was used to relay the stimulus signal to a Brüel + Kjær LDS PA100E Amplifier that drove a controlled 60 Hz vibratory stimulus along the larvae's dorsoventral axis (vertically). Two accelerometers (BU-21771, Knowles, Itasca, IL) were mounted to the head expander to monitor the vertical displacement of the plate. The output of the accelerometers was relayed through a custom accelerometer amplifier (EPL Engineering Core). A block diagram for the EPL Lateral Line Stimulator can be found here: https://www.masseyeandear.org/research/otolaryngology/eaton-peabody-laboratories/engineering-core.

## Mechanical overstimulation of lateral-line organs in free swimming larvae

At seven dpf, free-swimming zebrafish larvae were placed in untreated six-well plates (Corning, Cat# 3736; well diameter: 34.8 mm; total well volume: 16.8 ml) with 6 ml E3 per well, pre-warmed to 29 °C. Up to 15 larvae were placed in each well. Individual wells were sealed with Glad Press 'n Seal plastic food wrap prior to placing the lid on the plate. An additional metal plate was fitted to the bottom of the multi-well dish to fill a small gap between the bottoms of the wells and the head expander.

Mechanical water displacement exposures (stimulus parameters: 60 Hz, 40.3 ± 0.5 m/s²) were conducted at room temperature (22°C–24°C) up to 2 hr after dawn. The frequency selected for mechanical overexposure of lateral-line organs was based on previous studies showing 60 Hz to be within the optimal upper frequency range of mechanical sensitivity of superficial posterior lateral-line neuromasts (*Weeg et al., 2002*; *Trapani et al., 2009*, *Levi et al., 2015*). To confirm that 60 Hz was the optimal frequency to induce damage, we tested 45, 60, and 75 Hz at comparable intensities. We

observed at 75 Hz no apparent damage to lateral line neuromasts while 45 Hz at a comparable intensity proved toxic that is it was lethal to the fish.

Exposures consisted of 20 min of stimulation followed by a 10 -min break and 2 hr of uninterrupted stimulation. We also tested periodic exposures that consisted of a series of short pulses spanning 2 hr total: 2 20 min exposures each followed by 10 min of rest, followed by 30 min of stimulation, a 10 min break, and a final 20 min of stimulation. During the entire duration of exposure, unexposed control fish were kept in the same conditions as exposed fish i.e. placed in a multi-well dish and maintained in the same room as the exposure chamber. For experiments pharmacologically blocking glutamate uptake, fish were co-exposed to 10 µM DL-TBOA (Tocris; Cat. No.1223) + 0.1 % DMSO or 0.1 % DMSO alone. After exposure, larvae were either immediately fixed for histology, prepared for live imaging, or allowed to recover for up to 2 days in an incubator at 29 °C.

## Ablation of lateral-line organ with $CuSO_4$

Free-swimming larvae were exposed to freshly made 3 µM $CuSO_4$ solution in E3 for 1 hr, then rinsed and allowed to recover for 2 hr to ensure complete ablation of the lateral-line neuromasts. Neuromast ablation was confirmed by immunofluorescent labeling of hair cells. The effects of low-dose copper exposure are likely specific to lateral-line organs; a previous study in zebrafish determined exposure to low-dose $CuSO_4$ for 1 hr did not alter the acoustic escape response, which is similar to the fast start response we observed but evoked by higher frequency stimulation (100–500 Hz) of the anterior macula of inner ear (*Buck et al., 2012*).

## Fast-start escape response behavior assay

Images of larval swimming behavior (1000 frames per second) were acquired with an Edgertronic SC1 high-speed camera (Sanstreak Corp). Image acquisition began 10 s following stimulus onset. All subsequent analysis was performed using ImageJ. To track swimming behavior, images were initially stabilized using the Image Stabilizer Plugin. In stabilized images, the position of individual larval heads (located via the pigmented eyes) in each frame were tracked using the Manual Tracking Plugin. Larvae were tracked over 10 s (10,000 frames total) per trial. 'Fast start' responses—defined as a c-bend of the body occurring within 15 ms followed by a counter-bend— were identified manually.

## Whole-mount immunohistochemistry

For visualization of zebrafish lateral-line hair cells, neurons, and synapses: 7–9 dpf larvae were briefly sedated on ice, transferred to fixative (4 % paraformaldehyde, 1 % sucrose, 37.5 µM $CaCl_2$, 0.1 M phosphate buffer) in a flat-bottomed 2 ml Eppedorf tubes, and fixed for 5 hr at 4–8°C. Fixed larvae were permeabilized in ice-cold acetone for 5 minutes, then blocked in phosphate-buffered saline (PBS) with 2 % goat serum, 1 % bovine serum albumin (BSA), and 1 % DMSO for 2–4 hours at room temperature (RT; 22-24 °C). Larvae were incubated with primary antibodies diluted in PBS with 1 % BSA and 1 % DMSO overnight at 4–8°C, followed by several rinses in PBS/BSA/DMSO and incubation in diluted secondary antibodies conjugated to Pacific Blue (1:400), Alexa Fluor 488 (1:1000), Alexa Fluor 555 (1:1000), Alexa Fluor 647 (1:1000; Invitrogen), or DyLight 549 (1:1000; Vector Laboratories) for 2 hr at RT. In some experiments, fixed larvae were stained with 2.5 ug/ml 4',6-diamidino-2-phenylindole (DAPI; Invitrogen) diluted in PBS to label all cell nuclei. Larvae were mounted on glass slides with elvanol (13% w/v polyvinyl alcohol, 33% w/v glycerol, 1% w/v DABCO (1,4 diazobicylo[2,2,2] octane) in 0.2 M Tris, pH 8.5) and #1.5 cover slips.

For visualization of inflammation and macrophage recruitment: 7 dpf larvae were sedated on ice, transferred to 4 % paraformaldehyde fixative in PBS, then fixed overnight at 4°C–8°C. The next day larvae were rinsed in PBS and blocked in PBS with 5 % normal horse serum (NHS), 1 % DMSO, and 1 % Triton x-100 for 2 hr at RT. Larvae were incubated with primary antibodies diluted in PBS with 5 % NHS and 1 % Triton-x 100 overnight at RT, rinsed several times in PBS, then incubated in diluted secondary antibodies listed above for 2 hr at RT. Larvae were mounted on glass slides with glycerol/PBS (9:1); coverslips were sealed with clear nail polish.

## Primary antibodies

The following commercial antibodies were used in this study: GFP (1:500; Aves Labs, Inc; Cat# GFP-1020), Parvalbumin (1:2000; Thermo Fisher; Cat# PA1-933), Parvalbumin (1:2000; Abcam; Cat#

ab11427), Parvalbumin (1:500; Sigma-Aldrich; Cat# P3088), MAGUK (K28/86; 1:500; NeuroMab, UC Davis; Cat# 75–029), Otoferlin (1:500; Developmental Studies Hybridoma Bank/ HCS-1). Custom affinity-purified antibody generated against *Danio rerio* Ribeye b (mouse IgG2a; 1:2000; *Sheets et al., 2011*) was also used.

## Hair-cell labeling

To selectively label hair-cell nuclei, live zebrafish larvae were incubated with DAPI (5 mg/ml) diluted 1:2000 in E3 media for 4 min. Larvae were briefly rinsed three times in fresh E3 media, then immediately exposed to mechanical overstimulation.

Live imaging of FM1-43 did not provide the temporal resolution needed to compare relative uptake and fast recovery 0–4 hr following exposure. We therefore examined FM1-43 uptake using the fixable analogue. To label with FM1-43X (n-(3-triethylammoniumpropyl)–4-(4-(dibutylamino)-styryl) pyridinium dibromide; ThermoFisher), free-swimming larvae were exposed to FM 1–43 FX at 3 µM for 20 seconds, then rinsed three times in fresh E3 as previously described (*Toro et al., 2015*) and immediately fixed (4 % paraformaldehyde, 4 % sucrose, 150 µM $CaCl_2$, 0.1 M phosphate buffer). FM 1–43 FX mean signal intensity from maximum projection images was calculated using ImageJ as the integrated pixel intensity divided by the area of the neuromast region of interest. We verified that relative labeling of hair cells at 1–3 hours appeared comparable between live FM1-43 and FM 1–43 FX. We also verified loss of FM 1–43 FX uptake in larvae following brief treatment with 5 mM BAPTA to disrupt tip links (*Kindt et al., 2012*). Following fixation, stereocilia were labeled by incubation with phalloidin conjugated to Alexa Fluor 488 (Invitrogen) at 66 µM in PBS, washed, and mounted on slides with elvanol.

## EdU labeling and quantification

To label proliferating cells, we used the Click-iT EdU Cell Proliferation Kit for Imaging, Alexa Fluor 555 dye (Invitrogen). Following mechanical overstimulation, larvae were incubated in 500 µM EdU with 0.5 % DMSO in E3 for 4 hr at 28 °C then fixed in 4 % paraformaldehyde in PBS overnight at 4 °C. Larvae were washed in 3 % bovine serum albumin in PBS then permeabilized with 0.5 % Triton-X in PBS. GFP signal in Tg[tnks1bp1:GFP] fish was amplified using anti-GFP primary antibody (Aves), followed by a secondary antibody conjugated to Alexa Fluor 488. The EdU detection reaction was performed according to manufacturer guidelines; larvae were incubated in a reaction cocktail (4 mM CuSO4 and Alexa Fluor 555 azide in 1 X Click-iT reaction buffer with reaction buffer additive) for 1 hr at 25 °C. Larvae were washed, counterstained with Hoechst 33342, and mounted on slides with elvanol. Confocal images of neuromasts were acquired using an LSM 700 laser scanning confocal microscope with a 63 × 1.4 NA Plan-Apochromat oil-immersion objective (Carl Zeiss). The numbers of EdU+ cells per neuromast were quantified in ImageJ.

## Confocal imaging

Images of fixed samples were acquired using an LSM 700 laser scanning confocal microscope with a 63 × 1.4 NA Plan-Apochromat oil-immersion objective (Carl Zeiss). Confocal stacks were collected with a z step of 0.3 µm over 7–10 µm with pixel size of 100 nm (x-y image size 51 × 51 µm). Acquisition parameters were established using the brightest control specimen such that just a few pixels reached saturation in order to achieve the greatest dynamic range in our experiments. These parameters including gain, laser power, scan speed, dwell time, resolution, and zoom, were kept consistent between comparisons.

## Confocal image processing and analysis

All analysis was performed on blinded images. Digital images were processed using ImageJ software (*Schneider et al., 2012*). In order to quantitatively measure sizes and fluorescent intensities of puncta, raw images containing single immunolabel were subtracted for background using a 20-pixel rolling ball radius and whole neuromasts were delineated from Parvalbumin-labeled hair cells using the freehand selection and 'synchronize windows' tools. Puncta were defined as regions of immunolabel with pixel intensity above a determined threshold: threshold for Ribeye label was calculated using the Isodata algorithm (*Ridler and Calvard, 1978*) on maximum-intensity projections, threshold for MAGUK label was calculated as the product of 7 times the average pixel intensity of the whole neuromast region

in a maximum-intensity projection. Particle volume and intensity were measured using the 3D object counter (*Bolte and Cordelières, 2006*) using a lower threshold and a minimum size of 10 voxels. To normalize for differences in staining intensity across experimental trials, all volumes were divided by the median control volume in each trial for each individual channel. The number of particles above lower threshold was quantified using the ImageJ Maximum Finder plugin with a noise tolerance of 10 on maximum-intensity projections. Intact synapses were manually counted and defined as adjoining or overlapping maxima of Ribeye and MAGUK labels. The number of synapses per hair cell was approximated by dividing the number of intact synapses within a neuromast by the number of hair cells in the neuromast. Innervation of neuromast hair cells was quantified during blinded analysis by scrolling through confocal z-stacks of each neuromast (step size 0.3 μm) containing hair cell and afferent labeling and identifying hair cells that were not directly contacted by an afferent neuron i.e. no discernable space between the hair cell and the neurite. Hair cells that were identified as no longer innervated showed measurable neurite retraction; there was generally >0.5 μm distance between a retracted neurite and hair cell. Stereocilia length measurements were obtained in ImageJ from inter-polated 3D projections of z-stack images (step size 0.3 μm) containing phalloidin labeled hair bundles. Three independent measurements were obtained from the base to the tips of hair bundles at center of each neuromast, and the average length was calculated.

Quantitative data on macrophage response to mechanical injury were collected from the two caudal-most ('terminal') neuromasts. Confocal image stacks were obtained using a Zeiss LSM700 microscope and visualized using Volocity software. These image stacks were used to derive three metrics from each neuromast. First, the number of macrophages within 25 μm of a particular neuro-mast was determined by inscribing a circle of 25 μm radius, centered on the neuromast, and counting the number of macrophages that were either fully or partially enclosed by this circle. Next, the number of macrophages contacting a neuromast was determined by scrolling through the x-y planes of each image stack (1 μm interval between x-y planes, 15 μm total depth) and the counting macrophages that were in direct contact with Otoferlin-labeled hair cells. Finally, the number of macrophages that had internalized Otoferlin-labeled material (hair-cell debris) were counted and were assumed to reflect the number of phagocytic events. For each metric, the recorded number reflected the activity of a single macrophage, that is, a macrophage that made contacts with multiple hair cells and/or had internalized debris from several hair cells was still classified as a single 'event.'

Subsequent image processing for display within figures was performed using Photoshop and Illus-trator software (Adobe).

## Scanning electron microscopy

To image hair-cell bundles, zebrafish larvae were exposed to strong water wave stimulus, then anes-thetized in 0.12 % tricaine in E3 and immediately fixed in 2.5 % glutaraldehyde in 0.1 M sodium caco-dylate buffer (Electron Microscopy Sciences) supplemented with 2 mM $CaCl_2$. Larvae were shipped overnight in fixative, then most of the fixative (~90–95%) was removed, replaced with distilled water, and samples were stored at 4 C. Next, larvae were washed in distilled water (Gibco), dehydrated with an ascending series of ethanol, critical point dried from liquid $CO_2$ (Tousimis Aurosamdri 815), mounted on adhesive carbon tabs (Ted Pella), sputter coated with 5 nm of platinum (Leica EM ACE600), and imaged on Hitachi S-4700 scanning electron microscope. Kinocilia diameter measurements were performed using ImageJ.

## Statistical analysis

A hierarchical linear model analysis with fish used as random effects was used to compare each of the measures between the conditions and groups. Akaike Information Criterion (AIC) and Bayessian Infor-mation Criterion (BIC) were used to identify the best fitted covariance structure. Tukey's adjustment was used for the alpha level to avoid type I error inflation due to multiple comparisons. Estimated marginal mean differences and 95 % Confidence Intervals around them were explored and reported for quantification of effect size for group differences. Graphs for data visualization and additional statistical analyses were performed Prism 8 (Graphpad Software Inc). Mixed model analysis was used to compare time-series data. Statistical significance between synaptic ribbon and PSD volumes with was determined by Kruskal-Wallis test (one independent variable) or Mann–Whitney U test (one inde-pendent variable) and appropriate post-hoc tests. Based on the variance and effect sizes reported

in previous studies, the number of biological replicates were suitable to provide statistical power to avoid Type II error (*Sebe et al., 2017*; *Uribe et al., 2018*).

## Acknowledgements

This work was supported by the National Institute on Deafness and Other Communication Disorders R01DC016066 (LS), R01DC017166 (AAI), and R01DC006283 (MEW), Washington University Dept. of Otolaryngology (LS), and the Amelia Peabody Charitable Fund (LS). We thank Valentin Militchin (WashU) and Evan Foss (Mass Eye and Ear) for engineering support and Mark Rutherford for thoughtful feedback on the manuscript.

## Additional information

### Funding

| Funder | Grant reference number | Author |
| --- | --- | --- |
| National Institute on Deafness and Other Communication Disorders | R01DC016066 | Lavinia Sheets |
| National Institute on Deafness and Other Communication Disorders | R01DC017166 | Artur A Indzhykulian |
| National Institute on Deafness and Other Communication Disorders | R01DC006283 | Mark E Warchol |
| Washington University School of Medicine in St. Louis | | Lavinia Sheets |
| Amelia Peabody Charitable Fund | | Lavinia Sheets |

The funders had no role in study design, data collection and interpretation, or the decision to submit the work for publication.

### Author contributions

Melanie Holmgren, Conceptualization, Data curation, Formal analysis, Investigation, Methodology, Writing – review and editing; Michael E Ravicz, Methodology, Resources; Kenneth E Hancock, Software; Olga Strelkova, Data curation, Formal analysis; Dorina Kallogjeri, Formal analysis; Artur A Indzhykulian, Data curation, Formal analysis, Funding acquisition, Writing – review and editing; Mark E Warchol, Conceptualization, Data curation, Formal analysis, Funding acquisition, Writing – review and editing; Lavinia Sheets, Conceptualization, Data curation, Formal analysis, Funding acquisition, Investigation, Methodology, Project administration, Supervision, Writing - original draft

### Author ORCIDs

Melanie Holmgren [ORCID] http://orcid.org/0000-0002-2541-4854
Michael E Ravicz [ORCID] http://orcid.org/0000-0001-9978-3444
Artur A Indzhykulian [ORCID] http://orcid.org/0000-0002-2076-6818
Mark E Warchol [ORCID] http://orcid.org/0000-0002-0445-6318
Lavinia Sheets [ORCID] http://orcid.org/0000-0001-5231-2450

### Ethics

Ethics StatementThis study was performed with the approval of the Institutional Animal Care and Use Committee of Washington University School of Medicine in St. Louis (protocol no. 20–0158) and in accordance with NIH guidelines for use of zebrafish.

### Decision letter and Author response

Decision letter https://doi.org/10.7554/eLife.69264.sa1
Author response https://doi.org/10.7554/eLife.69264.sa2

## Additional files

### Supplementary files
• Transparent reporting form

### Data availability
All data generated or analyzed during this study are included in the manuscript and supporting files. Source data files have been provided for Figures 2, 3, 4, and 7.

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
