## [Decision Letter]

**Acceptance summary:**

Mechanical insults can lead to sensory hair cell death but the underlying mechanisms are not well understood. Here, Holmgren et al. describes for the first time, a mechanical hair-cell damage model in the zebrafish lateral line system, which allowed a closer examination of hair cell response to mechanical injury and recovery.

**Decision letter after peer review:**

[Editors’ note: the authors submitted for reconsideration following the decision after peer review. What follows is the decision letter after the first round of review.]

Thank you for submitting your work entitled "Mechanical overstimulation causes acute injury followed by fast recovery in lateral-line neuromasts of larval zebrafish" for consideration by *eLife*. Your article has been reviewed by 3 peer reviewers, and the evaluation has been overseen by a Reviewing Editor and a Senior Editor. The reviewers have opted to remain anonymous.

Our decision has been reached after consultation between the reviewers. Based on these discussions and the individual reviews below. Although all three reviewers agreed that the study was thorough but felt the findings do not provide sufficient advances to be considered further for publication in *eLife*.

*Reviewer #1:*

In the manuscript titled "Mechanical overstimulation causes acute injury followed by fast recovery in lateral-line neuromasts of larval zebrafish" by Holmgren et al., the authors develop a method to overstimulate hair cells and determine some of the consequences of this overstimulation. The overarching goal of this work is to develop a model for noise-induced hair-cell damage in the zebrafish. The authors use the lateral line for their studies and stimulate hair cells using an electrodynamic shaker which generate significant aqueous agitation. The authors demonstrate physical damage to hair cells of the lateral line that are dependent on position of the neuromast. The damage includes alteration of afferent synapses, afferent neurite retraction, limited damage to hair bundles and a decrease in mechanotransduction. After damage, they show macrophage recruitment and quick recovery of hair cell neuromasts, which is surprising.

The paper is interesting in that it brings a new capacity to the zebrafish animal model: mechanical overstimulation of the hair cell. Tempering this is a general feeling that the authors do not dig deep enough in the current form of the manuscript, but this could be remedied. More specifically, the authors are making a model in zebrafish for noise-induced damage, so they need to show that this model is similar to mammals in the way hair cells are damaged. This is done in the manuscript, but it is limited and should be expanded as suggested below.

– The authors use a vertically-oriented Brüel+Kjær LDS Vibrator to deliver a 60 Hz vibratory stimulus to damage lateral line hair cells. It is not made clear on why this frequency was selected. Did the authors choose this frequency because they screened a number of frequencies, and this is the one that did the most damage to hair cells or was it chosen for another reason? Or do all frequencies do the same amount of damage? The authors should screen a number of frequencies and choose the stimulus that does the most damage to hair cells. This would set the field in the best direction, should members of the community attempt this new technique. It is not necessary to repeat all of the experiments, but the authors should show which frequencies are best for inducing damage.

– The SEM images of the hair bundle are beautiful and do show damage to the hair bundle, but historically speaking older studies in mammals have shown that the actin core of the stereocilia is damaged. It would be critical to know if this was the case. Showing damage to the kinocilium and stereocilia splaying is a start, but readers of *eLife* would need to know if the actin cores are damaged. So, TEM should be used to find damage to the actin cores of stereocilia.

– I think the use of "Noise-exposed lateral line" as a term for mechanically overstimulated lateral line hair cells is not correct and could be misleading. The lateral line senses water motion, not sounds as the word noise would imply. Calling the stimulus "noise" should be removed throughout.

– Decreases in mechanotransduction are shown by dye entry. These results should be strengthened using microphonic potentials to determine the extent of damage. This experiment is not necessary but would improve the quality of the document.

– In figure 2, PSD labeling is not clear.

*Reviewer #2:*

Holmgren et al. describe the development of a model for hair cell noise damage using the zebrafish lateral line line system. Using an electrodynamic shaker, the authors induce quantifiable damage and death of hair cells after a two-hour treatment. They describe gross morphological changes of hair cells, changes in innervation and synapse distribution. In addition, they describe disruption of stereocilia and kinocilia, as well as reduced mechanotransduction-dependent uptake of FM1-43 dye. Damage is no longer detectable several hours after insult, demonstrating recovery.

1. While the findings are carefully measured and described the effects of insult on hair cells are relatively minor, with a change in hair cell number, extent of innervation or synapses per hair cell (Figures3 and 4) in the range of 10% reduction compared to control. One potential value of the model would be to use it to discover underlying pathways of damage or screen for potential therapeutics. However, with these modest changes it is not clear that there will be enough power to determine effects of potential interventions.

2. The most dramatic phenotype after shaking is a physical displacement of hair cells, described as disrupted morphology. However, it is not clear what the underlying cause of this change. Are only posterior neuromasts damaged in this way? Is it a wounding response as animals are exposed to an air interface during shaking? It is also not clear to what extent this displacement reveals more general principles of the effects of noise on hair cells. Additional discussion of underlying causes would be welcome.

3. Because afferent neurons innervate more than one neuromast and more than one hair cell per neuromast, measurements of innervation of neuromasts (Figure 3) or synapses per hair cell (Figure 4) cannot be assumed to be independent events. That is, changes in a single postsynaptic neuron may be reflected across multiple synapses, hair cells, and even neuromasts. This needs to be accounted for in experimental design for statistical analysis.

4. The SEM analysis provides compelling snapshots of apical damage but could be supplemented by quantitative analysis with antibody staining or transgenic lines where kinocilia are labeled. The amount of reduced FM1-43 labeling is one of the more dramatic effects of the shaking insult, suggesting widespread disruption to mechanotransduction that could be related to this apical damage. Further examination of the recovery of mechanotransduction would be interesting.

5. A previous publication by Uribe et al.2018 describes a somewhat similar shaking protocol with somewhat different results – more long-lasting changes in hair cell number, presynaptic changes in synapses, etc. It would be worth discussing potential differences across the two studies.

*Reviewer #3:*

Holmgren et al. describe a novel model of reversible mechanical damage to zebrafish neuromast hair cells. The authors demonstrate that when zebrafish are exposed to strong currents, neuromast morphology, hair cell number, innervation, and MET function suffer various types and degrees of damage, from which the NMs recover within 2 days. Additionally, they show macrophage recruitment to damaged neuromasts, where they may be phagocytosing synaptic debris. Based on various mechanistic and phenotypic commonalities (involvement of ROS, stereocilia and synapse phenotype), the authors argue that this model is a good approximation of noise-induced hair cell damage in mammals.

Overall impact:

This reviewer agrees that a "noise" damage model in the zebrafish would be a powerful tool to better understand the mechanisms underlying noise-induced hearing loss. However, due to various weaknesses of the data (detailed below), the main claims of the paper are not sufficiently supported. In addition, noise-induced hearing loss has been previously modeled in the zebrafish model. The present model, therefore, does not provide a significant methodological innovation. Based on this, and the fact that addressing all the concerns listed below likely exceeds the scope of a reasonable revision, this manuscript is believed to lack the impact and novelty to be recommended for publication in *eLife*.

– As the authors point out, zebrafish hair cells can be regenerated. With that in mind, and to make the relevance for mammalian hair cell repair clear, a clear distinction between mechanisms mediated by "repair" or "regeneration" needs to be made. The authors discuss that proliferative hair cell generation can be excluded based on the short time period, but suggest that transdifferentiation might be involved. Recovery of NM hair cell number occurs within the same 2 hour period in which NM morphology and hair cell function improved, making it difficult to determine the extent to which "regeneration" contributed to the recovery. The amount of transdifferentiation has to be shown experimentally (lineage tracing?).

– The classification of "normal" vs "disrupted" is vague and not quantitative. The examples shown in the paper seem to be quite clear-cut, but this reviewer doubts that was the case throughout all analyzed samples. Formulate clear benchmarks and criteria for the disrupted phenotype (even when blind analysis is performed).

– Sustained and periodic exposure: These two exposure protocols not only differ with respect to sustained vs periodic, they also differ in total exposure time (Figure 2B). This complicates the interpretation, especially considering the authors own finding that a pre-exposure is protective.

– The data on the mitochondrial ROS aspect seems not well integrated into the overall story.

– It is surprising that the hair bundle morphology was not assessed after recovery. This is crucial. Overall, it would be good to see some quantification of the SEM data, e.g. kinocilia length and number of splayed bundles.

– Behavioral recovery (measured as number of "fast start" responses) was also not assessed. This is essential for determining the functional relevance of the recovery.

– This reviewer is not yet convinced that this damage model displays enough commonalities to mammalian noise damage to justify the ubiquitous use of the term "noise" throughout the manuscript. It would be more prudent to use a more careful term along the lines of "mechanical overstimulation-induced damage".

– Overall, there was a lack of experimental and analysis detail in the Results section. For example, how was afferent innervation quantified? Just counting GFP labeled contacts to hair cells? There was also inconsistency in the use of two variations of the mechanical damage protocol, the time points at which repair was assessed, and whether the damage was quantified in all neuromasts or in normal vs. disrupted neuromasts separately, making the data difficult to interpret.

[Editors’ note: further revisions were suggested prior to acceptance, as described below.]

Thank you for resubmitting your work entitled "Mechanical overstimulation causes acute injury and synapse loss followed by fast recovery in lateral-line neuromasts of larval zebrafish" for further consideration by *eLife*. Your revised article has been evaluated by Didier Stainier (Senior Editor) and a Reviewing Editor.

All three reviewers appreciated the fact that you have made a good effort in revising the manuscript. However, the story has also changed from a noise-induced damage to a mechanical over-stimulation paradigm. While the Ihfp15b mutant results are interesting, it is not entirely clear how these results are applicable to noise-induced hearing loss in mammals. A point that could be better addressed in the Discussion. After an extensive discussion with the three reviewers, they have consolidated their main suggestions as follow:

Essential revisions

1) It would be important to determine whether the hair cells with severe bundle damage recover their bundle morphology or are replaced by transdifferentiation of SCs.

2) Statistical analyses of results should be incorporated per comments from Reviewer#2. It is important to know if the conclusions hold up across total neuromasts and fish. In this regard, this is a reference suggested by the Reviewer: Aarts E, Verhage M, Veenvliet JV, Dolan CV, van der Sluis S. A solution to dependency: using multilevel analysis to accommodate nested data. Nat Neurosci. 2014 Apr;17(4):491-6.

*Reviewer #1:*

The authors embark on a comprehensive characterization of damages suffered by the zebrafish lateral line organ in response to mechanical overstimulation. The ultimate goal is to establish this as a novel model for noise-induced hearing loss. The study describes significant overlap in the damage characteristics between mammals and fish, justifying this protocol as a (limited) but very useful model (due to the well-known advantages of the zebrafish system) to study the basic mechanisms of hair cell damage incurred by mechanical overstimulation.

The revision includes some valuable additions in response to the previous review, but many suggestions were deemed intractable in the given time period. The manuscript is therefore improved, but not to a level that this reviewer judges had hoped for.

– The quantification of changes in SEMs of hair bundle morphology following damage was a significant improvement from the original submission, but imaging or quantification of recovery, which would be an important addition to the manuscript, is still lacking. It would also be nice to see a quantification of splayed bundles and kinocilia length before and after recovery. The confocal images demonstrating the changes to the kinocilium following damage were a nice addition and might provide an easier way to quantify changes during the recovery process.

– The authors decided that further investigation into the distinction of repair vs recovery (beyond SC EdU incorporation) is outside the scope of this study, but this reviewer thinks that it should have been further explored. The authors claim that they only see a modest increase in SC EdU incorporation (>0.5 SC/neuromast) following injury, but the decrease in HC number is also small (looks to be <1 HC/neuromast from graph). Directly addressing transdifferentiation of SCs through lineage tracing techniques would be informative.

*Reviewer #2:*

I appreciate the extensive revisions Holmgren et al. have made to their previous manuscript, addressing many of the criticisms of the previous review. They include a clever new experiment using lhfpl5b mutants that have mechanosensory deficits specific to lateral line hair cells. Using these mutants, they demonstrate that the morphological disruption they observe after shaker treatment occurs in these mutants to the same degree as in wt fish.

However, this result does call into question whether the model is a functional one for noise damage. If mechanotransduction is not necessary, are the phenotypes observed really excitotoxicity? Do the reported changes in innervation and synapses also not require functional mechanotransduction?

I fear I was not clear enough on my previous point (#3 in response to reviewer 2):

Because afferent neurons innervate more than one neuromast and more than one hair cell per neuromast, measurements of innervation of neuromasts (Figure 3) or synapses per hair cell (Figure 4) cannot be assumed to be independent events. That is, changes in a single postsynaptic neuron may be reflected across multiple synapses, hair cells, and even neuromasts. This needs to be accounted for in experimental design for statistical analysis.

As measurements of innervation or synapse density are not independent, they cannot be treated as distinct instances (n's) in the statistical analysis. In these experiments the independent instances are going to be the number of fish, not hair cells. This issue extends to analysis in Figures7, 8 as well.

I am somewhat confused about the conclusions from figure 4. Setting aside the issue of independence, the differences in synapses per hair cell are variably measured as significant or insignificant, suggesting that the analysis is underpowered. In addition, shouldn't the relevant comparisons in Figure 4E be between DMSA and TBOE, that is comparing each of control, shaken normal and shaken disrupted to the corresponding drug treatment?

*Reviewer #3:*

The goal of the manuscript is to present a new paradigm for damaging hair cells of the zebrafish lateral line using vibrational stimuli. A successful method would be important to the community as this goal has been difficult to achieve, and it would open up new avenues of research. The authors addressed some of my concerns from the last review but not all. The experiments they did perform are a screen of frequencies that damage the hair cells of the lateral line and improved the quality of an image (Figure 4). This is appreciated especially during the time of Covid.

They argue that 2 suggested experiments are not necessary: microphonic potentials and damage to the core of actin of stereocilia. I feel that they would improve the paper's quality, but I see their point about not including the microphonic potentials; however, they should at some level look at the damage to the actin core. This could be by TEM as suggested or it could be by phalloidin labeling. Both of these methods can be used to see gaps in the core of actin in mouse hair cells and maybe those of zebrafish.

Another issue is the use of the word "current". According to the dictionary (OED) current is defined as "That which runs or flows, a stream; spec. a portion of a body of water, or of air, etc. moving in a definite direction." This is the way I see the word used in the scientific literature as well. I don't think what the authors are providing is a current. They should use a different term, perhaps, "vibratory stimulus" or "water stimulus."

---

## [Author Response]

[Editors’ note: the authors resubmitted a revised version of the paper for consideration. What follows is the authors’ response to the first round of review.]

Reviewer #1:In the manuscript titled "Mechanical overstimulation causes acute injury followed by fast recovery in lateral-line neuromasts of larval zebrafish" by Holmgren et al., the authors develop a method to overstimulate hair cells and determine some of the consequences of this overstimulation. The overarching goal of this work is to develop a model for noise-induced hair-cell damage in the zebrafish. The authors use the lateral line for their studies and stimulate hair cells using an electrodynamic shaker which generate significant aqueous agitation. The authors demonstrate physical damage to hair cells of the lateral line that are dependent on position of the neuromast. The damage includes alteration of afferent synapses, afferent neurite retraction, limited damage to hair bundles and a decrease in mechanotransduction. After damage, they show macrophage recruitment and quick recovery of hair cell neuromasts, which is surprising.The paper is interesting in that it brings a new capacity to the zebrafish animal model: mechanical overstimulation of the hair cell. Tempering this is a general feeling that the authors do not dig deep enough in the current form of the manuscript, but this could be remedied. More specifically, the authors are making a model in zebrafish for noise-induced damage, so they need to show that this model is similar to mammals in the way hair cells are damaged. This is done in the manuscript, but it is limited and should be expanded as suggested below.– The authors use a vertically-oriented Brüel+Kjær LDS Vibrator to deliver a 60 Hz vibratory stimulus to damage lateral line hair cells. It is not made clear on why this frequency was selected. Did the authors choose this frequency because they screened a number of frequencies, and this is the one that did the most damage to hair cells or was it chosen for another reason? Or do all frequencies do the same amount of damage? The authors should screen a number of frequencies and choose the stimulus that does the most damage to hair cells. This would set the field in the best direction, should members of the community attempt this new technique. It is not necessary to repeat all of the experiments, but the authors should show which frequencies are best for inducing damage.

The frequency selected for mechanical overexposure of lateral-line organs was based on previous studies showing 60 Hz to be within the optimal upper frequency range of mechanical sensitivity of superficial posterior lateral-line neuromasts, with maximal response between 10-60 Hz, but a suboptimal frequency for hair cells of the anterior macula in the ear (Weeg and Bass 2002, Trapani et al., 2009, Levi et al., 2015). To confirm that 60 Hz was the optimal frequency to induce damage, we tested 45, 60, and 75 Hz at comparable intensities. We observed at 75 Hz no apparent damage to lateral line neuromasts while 45 Hz at a comparable intensity proved toxic i.e. it was lethal to the fish. We have updated the Results and Method Details to include our rationale for choosing 60 Hz**.**

– The SEM images of the hair bundle are beautiful and do show damage to the hair bundle, but historically speaking older studies in mammals have shown that the actin core of the stereocilia is damaged. It would be critical to know if this was the case. Showing damage to the kinocilium and stereocilia splaying is a start, but readers of eLife would need to know if the actin cores are damaged. So, TEM should be used to find damage to the actin cores of stereocilia.

Our main goal of this initial manuscript was to survey morphological and functional changes in mechanically injured lateral line organs with an emphasis on inflammation and synapse loss. We agree TEM studies showing damage to the actin core of the stereocilia will be important to determine whether mechanical damage to neuromast hair bundles fully mimics mammalian stereocilia damage, but these experiments will require significant time to perform and optimize. We have expanded our analysis of hair-bundle morphology in this study and intend to pursue deeper analysis of hair bundle damage, i.e. examination of the stereocilia actin core, in future follow-up studies.

– I think the use of "Noise-exposed lateral line" as a term for mechanically overstimulated lateral line hair cells is not correct and could be misleading. The lateral line senses water motion, not sounds as the word noise would imply. Calling the stimulus "noise" should be removed throughout.

We have removed the term “noise” throughout the manuscript and replaced it with either “strong water current stimulus” or “mechanical overstimulation” where appropriate.

– Decreases in mechanotransduction are shown by dye entry. These results should be strengthened using microphonic potentials to determine the extent of damage. This experiment is not necessary but would improve the quality of the document.

While we agree that microphonic recordings would provide further support for reduced mechanotransduction, quantitative FM1-43 uptake in zebrafish lateral line hair cells is a well-established proxy for microphonic measurements. In a previous study using the same protocol utilized in our manuscript, FM1-43 labeling intensity was shown to directly correspond with microphonic amplitude (Toro et al., 2015). Moreover, the fixable analogue of FM1-43 (FM143FX) gave us comparable relative measurements of uptake as live FM1-43 and provided the additional advantage of high temporal resolution and the ability to simultaneously assay entire cohorts of control and overstimulated fish (which is not possible with microphonic measurements or live FM1-43 imaging), as we could expose groups of fish briefly to the dye at determined time intervals following overstimulation, then immediately place in fixative.

– In figure 2, PSD labeling is not clear.

We assume the reviewer meant PSD labeling in Figure 4 and we agree it is difficult to discern. We have changed the hair-cell label from gray to blue in the images so that the green PSD labeling is clear.

Reviewer #2:Holmgren et al. describe the development of a model for hair cell noise damage using the zebrafish lateral line line system. Using an electrodynamic shaker, the authors induce quantifiable damage and death of hair cells after a two-hour treatment. They describe gross morphological changes of hair cells, changes in innervation and synapse distribution. In addition, they describe disruption of stereocilia and kinocilia, as well as reduced mechanotransduction-dependent uptake of FM1-43 dye. Damage is no longer detectable several hours after insult, demonstrating recovery.1. While the findings are carefully measured and described the effects of insult on hair cells are relatively minor, with a change in hair cell number, extent of innervation or synapses per hair cell (Figures3 and 4) in the range of 10% reduction compared to control. One potential value of the model would be to use it to discover underlying pathways of damage or screen for potential therapeutics. However, with these modest changes it is not clear that there will be enough power to determine effects of potential interventions.

One advantage of the zebrafish model is the ability to overstimulate large cohorts of larvae, thereby providing enough power to uncover modest but significant changes resulting from moderate damage to hair cells. While not as well suited for unbiased large-scale screens of therapeutics, our overexposure protocol provides the opportunity to determine the role of specific cellular pathways (e.g. metabolic stress, inflammation, and glutamate excitotoxicity) in hair-cell damage and synapse loss following mechanically-induced damage via genetic or pharmacological manipulation of these pathways. Additionally, as the hair cell synapses fully repair following stimulus-induced loss, the zebrafish model has the potential for identifying novel pathways for repair through transcriptomic profiling (for an example, see Mattern et al., Front. Cell Dev. Biol., 2018). Cumulatively, these future experimental directions will provide important mechanistic information that could be used toward the development of targeted therapeutic interventions.

2. The most dramatic phenotype after shaking is a physical displacement of hair cells, described as disrupted morphology. However, it is not clear what the underlying cause of this change. Are only posterior neuromasts damaged in this way? Is it a wounding response as animals are exposed to an air interface during shaking? It is also not clear to what extent this displacement reveals more general principles of the effects of noise on hair cells. Additional discussion of underlying causes would be welcome.

We agree that the underlying causes of the physical displacement of posterior lateral-line neuromasts warranted further investigation and we have expanded appropriate sections of the results. To determine if excessive hair-cell activity plays a role in the displacement of neuromasts we have exposed *lhfpl5b* mutant—fish that have intact hair cell function in the ear, but no mechanotransduction in hair cells of the lateral line—to mechanical overstimulation. We observed comparable disruption of neuromasts lacking mechanotransduction, supporting that displacement of lateral-line hair cells is due to mechanical damage and does not require intact mechnotransduction. Further, when examining the adjacent supporting cells in disrupted neuromasts, we observed they are similarly displaced and elongated. We conclude that observed disruption of hair cells is a consequence of mechanical displacement of the entire neuromast organ. We have added additional discussion of this phenomenon to the Results and Discussion sections of the manuscript.

3. Because afferent neurons innervate more than one neuromast and more than one hair cell per neuromast, measurements of innervation of neuromasts (Figure 3) or synapses per hair cell (Figure 4) cannot be assumed to be independent events. That is, changes in a single postsynaptic neuron may be reflected across multiple synapses, hair cells, and even neuromasts. This needs to be accounted for in experimental design for statistical analysis.

We agree that changes in single postsynaptic neurons, which innervate groups of hair cells of the same polarity within a neuromast, could be reflected across multiple synapses. Additionally, it is plausable that excitotoxic events at the postsynapse, while not contributing to apparent neurite retraction, could be contributing to synapse loss across multiple innervated hair cells. We have updated the manuscript to reflect the potential contribution of postsynaptic signaling to synapse loss and added experiments pharmacologically blocking glutamate uptake.

4. The SEM analysis provides compelling snapshots of apical damage but could be supplemented by quantitative analysis with antibody staining or transgenic lines where kinocilia are labeled. The amount of reduced FM1-43 labeling is one of the more dramatic effects of the shaking insult, suggesting widespread disruption to mechanotransduction that could be related to this apical damage. Further examination of the recovery of mechanotransduction would be interesting.

To supplement the SEM snapshots of severe apical damage, we have expanded the SEM image analysis with quantitative data on kinocilia morphology. We have also added confocal images of hair bundles using antibody labeling of acetylated tubulin in a transgenic line expressing β-actin-GFP in hair cells. We agree that correlative studies of mechanotransduction recovery relative to hair-bundle morphology would be interesting, and we intend to examine this question in a future follow-up study.

5. A previous publication by Uribe et al.2018 describes a somewhat similar shaking protocol with somewhat different results – more long-lasting changes in hair cell number, presynaptic changes in synapses, etc. It would be worth discussing potential differences across the two studies.

We agree we did not adequately address the considerable differences between our mechanical damage protocol for the zebrafish lateral line and the damage protocol described by Uribe et al., 2018. We have provided a more direct comparison in the Results section and addressed the differences in our protocols in-depth in the Discussion section.

Our damage protocol uses a stimulus within the known frequency range of lateral-line hair cells (60 Hz) that is applied to free-swimming larvae and evokes a behaviorally relevant response (fast start response). The damage is observable immediately following noise exposure, is specific to posterior lateral-line neuromasts, and appears to be rapidly repaired. Some features of the damage we observe—reduced mechanotransduction and hair-cell synapse loss—may correspond to mechanically induced damage of hair cell organs in other species. Notably, hair cell synapse loss in seemingly intact neuromasts is exacerbated by pharmacologically blocking synaptic glutamate clearance, supporting that the 60 Hz frequency stimulus is overstimulating neuromast hair cells directly and suggesting that the mechanism of synapse loss may be similar to inner hair cell synapse loss reported in mice following moderate noise exposures.

By contrast, the damage protocol published by Uribe et al. used ultrasonic transducers (40-kHz) to generate small, localized shock waves rather than directly stimulate neuromast hair cells. The damaged they reported—delayed hair-cell death and modest synapse loss with no effect on hair-cell mechanotransduction—was not apparent until 48 hours following exposure and not specific to the lateral-line organ. Some of the features of the damage they observed—delayed onset apoptosis and hair-cell death—may correspond to damage reported in mice following blast injuries.

Reviewer #3:Holmgren et al. describe a novel model of reversible mechanical damage to zebrafish neuromast hair cells. The authors demonstrate that when zebrafish are exposed to strong currents, neuromast morphology, hair cell number, innervation, and MET function suffer various types and degrees of damage, from which the NMs recover within 2 days. Additionally, they show macrophage recruitment to damaged neuromasts, where they may be phagocytosing synaptic debris. Based on various mechanistic and phenotypic commonalities (involvement of ROS, stereocilia and synapse phenotype), the authors argue that this model is a good approximation of noise-induced hair cell damage in mammals.Overall impact:This reviewer agrees that a "noise" damage model in the zebrafish would be a powerful tool to better understand the mechanisms underlying noise-induced hearing loss. However, due to various weaknesses of the data (detailed below), the main claims of the paper are not sufficiently supported. In addition, noise-induced hearing loss has been previously modeled in the zebrafish model. The present model, therefore, does not provide a significant methodological innovation. Based on this, and the fact that addressing all the concerns listed below likely exceeds the scope of a reasonable revision, this manuscript is believed to lack the impact and novelty to be recommended for publication in eLife.– As the authors point out, zebrafish hair cells can be regenerated. With that in mind, and to make the relevance for mammalian hair cell repair clear, a clear distinction between mechanisms mediated by "repair" or "regeneration" needs to be made. The authors discuss that proliferative hair cell generation can be excluded based on the short time period, but suggest that transdifferentiation might be involved. Recovery of NM hair cell number occurs within the same 2 hour period in which NM morphology and hair cell function improved, making it difficult to determine the extent to which "regeneration" contributed to the recovery. The amount of transdifferentiation has to be shown experimentally (lineage tracing?).

We agree that the distinction between "repair" and "regeneration" needs to be made when discussing this model of mechanical damage to zebrafish hair cell organs. We have tried to clarify that most of what we observe regarding recovery—restoration of neuromast shape, mechanostransduction, afferent contacts, and synapse number —reflect mechanisms of repair following mechanical damage (and, in the case of synapse loss, overstimulation) rather than regeneration. However, one feature of damage that may reflect rapid regeneration is restoration of hair cells number following mechanical injury. To experimentally determine whether proliferation contributed to hair cell generation, we assessed the incorporation of the thymidine analog EdU during a 4 hour recovery following mechanical overexposure in a transgenic line expressing GFP in neuromast supporting cells and observe a modest but not statistically significant increase in the number of proliferating supporting cells in neuromasts exposed to strong current stimulus, suggesting recovery of lost hair cells is not primarily due to renewed proliferation.

The number of hair cells that are lost and recover within several hours are low, i.e., typically ~1 hair cell/neuromast. We observed this consistently in all of our experiments, but the mechanisms responsible are not clear. Based on previous studies of hair cell regeneration in the lateral line, the recovery time appears too rapid to be caused by renewed proliferation, a notion that is further supported by our Edu studies. On the other hand, it is possible that a few supporting cells may undergo the initial phases of phenotypic change into hair cells during this short time period, and we speculate that such transdifferentiation may be responsible for the observed recovery. We should emphasize that this is a new observation and, at present, we do not fully understand the underlying mechanism. However, the focus of the present study is on mechanical damage, synaptic loss, and subsequent repair. We believe that it is important to report our consistent findings of low level hair cell loss and recovery, but a detailed characterization of the mechanism would require considerable effort and would best be the topic of a future study.

– The classification of "normal" vs "disrupted" is vague and not quantitative. The examples shown in the paper seem to be quite clear-cut, but this reviewer doubts that was the case throughout all analyzed samples. Formulate clear benchmarks and criteria for the disrupted phenotype (even when blind analysis is performed).

We have defined measurable criteria for "normal" vs "disrupted" neuromasts that we have added to the Method Details section: “We defined exposed neuromast morphology as “normal” when hair cells appeared radially organized with a relatively uniform shape and size, with ≤7 µm difference observed when comparing the lengths from apex to base of an opposing pair of anterior/posterior hair cells. Length was measured from a fixed point at the center of the hair bundle to the basolateral end of each opposing hair cell. We defined neuromasts as “disrupted” when hair cells appeared elongated and displaced to one side, with >7 µm difference observed when comparing the lengths of an opposing pair of anterior/posterior hair cells. Generally, the apical ends of the hair cells were displaced posteriorly, with the basolateral ends oriented anteriorly.”

– Sustained and periodic exposure: These two exposure protocols not only differ with respect to sustained vs periodic, they also differ in total exposure time (Figure 2B). This complicates the interpretation, especially considering the authors own finding that a pre-exposure is protective.

To clarify—pre-exposure was not protective to hair-cell survival. Rather, in preliminary experiments, pre-exposure appeared to reduce larval mortality, and we have clarified that observation in the text of the Results and the Methods Details sections. We agree with the reviewer that comparing the two protocols based on differences in time distribution is complicated in that they also differ in total exposure time. For the purpose of clarity, we now focus on the sustained exposure in the main figures and created supplemental figures for the reduced damage still observed using periodic exposure, specifying that reduced damage may be the result of periodic time distribution of stimulus and/or less cumulative time exposed to the stimulus.

– The data on the mitochondrial ROS aspect seems not well integrated into the overall story.

We agree that the ROS story was not well integrated and incomplete. We have removed the data describing *mpv17-/-* mutants and mitochondrial disfunction from this manuscript. A more comprehensive report of *mpv17-/-* mutant mitochondrial function and morphological analysis of neuromasts following noise exposure will be described in a follow-up manuscript.

– It is surprising that the hair bundle morphology was not assessed after recovery. This is crucial. Overall, it would be good to see some quantification of the SEM data, e.g. kinocilia length and number of splayed bundles.

We have expanded the SEM image analysis to quantitatively access kinocilia morphology following exposure. We agree that assessment of recovery using live-imaging of hair bundles paired with subsequent SEM analysis will be informative, and we intend to perform those experiments in a future study.

– Behavioral recovery (measured as number of "fast start" responses) was also not assessed. This is essential for determining the functional relevance of the recovery.

We attempted to measure behavior recovery of lateral-line function by measuring “fast-start” responses immediately and several hours after recovery, and discovered that i) strong water current provided stimulation that was too intense to reveal subtle behavioral changes following lateral-line damage and recovery, and ii) when testing larvae immediately following sustained strong current exposures, it was difficult to discern if fewer “fast-start” responses were due to lateral-line organ damage or larval fatigue.

We agree that behavioral recovery is important to assay but acknowledge assessing lateral-line mediated behavior following mechanical damage will require a more sensitive testing paradigm that stimulates the lateral-line sensory organ with a relatively gentile, calibrated water flow stimulus. We are currently performing a follow-up study to this paper using a testing paradigm developed by a postdoctoral associate in our lab (Kyle Newton) that analyses subtle changes in larval orientation to water flow (rheotaxis) mediated by the lateral-line organ. Using this behavior paradigm, we will directly correlate morphological and functional recovery over time.

– This reviewer is not yet convinced that this damage model displays enough commonalities to mammalian noise damage to justify the ubiquitous use of the term "noise" throughout the manuscript. It would be more prudent to use a more careful term along the lines of "mechanical overstimulation-induced damage".

We have removed the term “noise” throughout the manuscript and replaced it with either “strong water current stimulus” or “mechanical overstimulation” where appropriate.

– Overall, there was a lack of experimental and analysis detail in the Results section. For example, how was afferent innervation quantified? Just counting GFP labeled contacts to hair cells?

Innervation of neuromast hair cells was quantified during blinded analysis by scrolling through confocal z-stacks of each neuromast (step size 0.3 µm) containing hair cell and afferent labeling and identifying hair cells that were not directly contacted by an afferent neuron i.e. no discernable space between the hair cell and the neurite. Hair cells that were identified as no longer innervated showed measurable neurite retraction; there was generally >0.5 µm distance between a retracted neurite and hair cell. We have added this information to the Methods Detail section.

There was also inconsistency in the use of two variations of the mechanical damage protocol, the time points at which repair was assessed, and whether the damage was quantified in all neuromasts or in normal vs. disrupted neuromasts separately, making the data difficult to interpret.

We have revised our figure legends to clearly indicate when we are assessing damage in all exposed neuromasts (pooled) to control vs. comparative analysis of normal vs. disrupted neuromasts relative to control. In addition, we now focus on the sustained exposure in the main figures, which was the exposure protocol used for the time points in which repair and recovery were assessed.

[Editors’ note: what follows is the authors’ response to the second round of review.]

All three reviewers appreciated the fact that you have made a good effort in revising the manuscript. However, the story has also changed from a noise-induced damage to a mechanical over-stimulation paradigm. While the Ihfp15b mutant results are interesting, it is not entirely clear how these results are applicable to noise-induced hearing loss in mammals. A point that could be better addressed in the Discussion. After an extensive discussion with the three reviewers, they have consolidated their main suggestions as follow:Essential revisions1) It would be important to determine whether the hair cells with severe bundle damage recover their bundle morphology or are replaced by transdifferentiation of SCs.

Our revised manuscript provides data that indicate disrupted neuromasts, which appear to correspond with damaged hair bundles in our SEM imaging experiments, are repaired rather than replaced following mechanical damage.

We performed time series experiments examining mechanotransduction and neuromast morphology 0-4 hours following strong water wave exposure, which are now reported in the revised Figure 10. We observed hair-cell mechanotransduction, as measured with FM1-43X uptake, was significantly reduced immediately following exposure. However, uptake of FM1-43X began to recover within 30 minutes and was fully restored after several hours. Notably, reduced FM1-43X uptake did not appear to correlate with stereocilia length. Recovery of FM1-43X fluorescence instead coincided with a relative decrease in the number of neuromasts that appeared “disrupted”, suggesting restoration of overall neuromast morphology. In addition, we expanded our evaluation of cell proliferation following 4 hours recovery and observed no significant increase in supporting cell EdU labeling in exposed neuromasts (Figure 11). Together, these observations suggest that neuromast hair cells disrupted by strong mechanical stimuli are generally repaired rather than replaced.

2) Statistical analyses of results should be incorporated per comments from Reviewer#2. It is important to know if the conclusions hold up across total neuromasts and fish. In this regard, this is a reference suggested by the Reviewer: Aarts E, Verhage M, Veenvliet JV, Dolan CV, van der Sluis S. A solution to dependency: using multilevel analysis to accommodate nested data. Nat Neurosci. 2014 Apr;17(4):491-6.

To accommodate nested data and ensure statistical rigor, our data has been reanalyzed by Dr. Dorina Kallogjeri, a research statistician affiliated with our department. Using an unconditional model, she determined ~35% of variability in hair cell number ~25% variability in innervation could be accounted for by the fish in the experiment. She therefore used a multilevel model to test statistical differences between conditions and groups. Tukey’s adjustment was used for the α level to avoid type I error inflation due to multiple comparisons.

In addition to updating our statistics and figures in the manuscript, we have included our raw data and her analysis in our resubmission.

Reviewer #1:The authors embark on a comprehensive characterization of damages suffered by the zebrafish lateral line organ in response to mechanical overstimulation. The ultimate goal is to establish this as a novel model for noise-induced hearing loss. The study describes significant overlap in the damage characteristics between mammals and fish, justifying this protocol as a (limited) but very useful model (due to the well-known advantages of the zebrafish system) to study the basic mechanisms of hair cell damage incurred by mechanical overstimulation.The revision includes some valuable additions in response to the previous review, but many suggestions were deemed intractable in the given time period. The manuscript is therefore improved, but not to a level that this reviewer judges had hoped for.– The quantification of changes in SEMs of hair bundle morphology following damage was a significant improvement from the original submission, but imaging or quantification of recovery, which would be an important addition to the manuscript, is still lacking. It would also be nice to see a quantification of splayed bundles and kinocilia length before and after recovery. The confocal images demonstrating the changes to the kinocilium following damage were a nice addition and might provide an easier way to quantify changes during the recovery process.

Quantification of phalloidin-labeled hair bundle length during recovery and in relation to FM1-43 uptake is now reported in revised Figure 10.

– The authors decided that further investigation into the distinction of repair vs recovery (beyond SC EdU incorporation) is outside the scope of this study, but this reviewer thinks that it should have been further explored. The authors claim that they only see a modest increase in SC EdU incorporation (>0.5 SC/neuromast) following injury, but the decrease in HC number is also small (looks to be <1 HC/neuromast from graph). Directly addressing transdifferentiation of SCs through lineage tracing techniques would be informative.

It is notable that prior studies of hair cell regeneration in the zebrafish lateral line suggest that the underlying cellular mechanisms are slightly different from those that occur in the inner ears of other nonmammalian vertebrates. The ears of birds and amphibians can generate replacement hair cells either by asymmetric division of supporting cells or by nonmitotic transdifferentiation of supporting cells (Robertson et al., 2004). In contrast, regeneration of zebrafish lateral line hair cells appears to occur via the symmetric division of supporting cells, and no prior studies, to our knowledge, have shown evidence for either transdifferentiation or production by asymmetric proliferation.

To address the issue of hair cell addition, we examined supporting cell proliferation (via EdU labeling) along with cell fate using a transgenic line that expressed a stable fluorophore (GFP) in supporting cells. In these fish, transdifferentiation of supporting cells should result in the colocalization of GFP with the hair-cell marker otoferlin, either immediately after or at short time intervals after mechanical injury. It should be noted that this method will label *both* hair cells that arise from transdifferentiation as well as newly-differentiated hair cells that were created by the proliferation of (GFP-expressing) supporting cells. In mechanically-damaged fish, we observed such GFP colocalization within 1-2 hair cells in a small fraction of neuromasts (3/109 neuromasts; n=41 fish, examined at 0-2 hr after exposure). However, we also observed similar colocalization in unexposed (control) fish (1/49 neuromasts, n=24 fish), suggesting that addition of new hair cells (by whatever mechanism) is similar in both damaged and control fish. In addition, supporting cell proliferation (as quantified by EdU labeling; Figure 11) did not significantly differ in mechanically injured vs. control fish. Together, these observations suggest that the mechanical trauma did not affect the rate at which new hair cells are generated. Finally, since hair cells in zebrafish neuromasts undergo a slow rate of turnover (e.g., Williams and Holder, 2000) it is expected that a low level of hair cell addition will continuously occur in control and mechanically overstimulated larvae.

Reviewer #2:I appreciate the extensive revisions Holmgren et al. have made to their previous manuscript, addressing many of the criticisms of the previous review. They include a clever new experiment using lhfpl5b mutants that have mechanosensory deficits specific to lateral line hair cells. Using these mutants, they demonstrate that the morphological disruption they observe after shaker treatment occurs in these mutants to the same degree as in wt fish.However, this result does call into question whether the model is a functional one for noise damage. If mechanotransduction is not necessary, are the phenotypes observed really excitotoxicity? Do the reported changes in innervation and synapses also not require functional mechanotransduction?

We believe the data showing significant synapse loss in neuromasts with overall intact morphology (i.e. “normal”) and our observation that blocking glutamate uptake significantly worsens synapse loss specifically in “normal” neuromasts provide strong evidence that synapse loss is a consequence of excitotoxicity.

I fear I was not clear enough on my previous point (#3 in response to reviewer 2):Because afferent neurons innervate more than one neuromast and more than one hair cell per neuromast, measurements of innervation of neuromasts (Figure 3) or synapses per hair cell (Figure 4) cannot be assumed to be independent events. That is, changes in a single postsynaptic neuron may be reflected across multiple synapses, hair cells, and even neuromasts. This needs to be accounted for in experimental design for statistical analysis.As measurements of innervation or synapse density are not independent, they cannot be treated as distinct instances (n's) in the statistical analysis. In these experiments the independent instances are going to be the number of fish, not hair cells. This issue extends to analysis in Figures7, 8 as well.

To accommodate nested data and ensure statistical rigor, we used a multilevel model to test statistical differences between groups.

I am somewhat confused about the conclusions from figure 4. Setting aside the issue of independence, the differences in synapses per hair cell are variably measured as significant or insignificant, suggesting that the analysis is underpowered. In addition, shouldn't the relevant comparisons in Figure 4E be between DMSA and TBOE, that is comparing each of control, shaken normal and shaken disrupted to the corresponding drug treatment?

We have included the appropriate comparisons in the updated Figure 4.

Reviewer #3:The goal of the manuscript is to present a new paradigm for damaging hair cells of the zebrafish lateral line using vibrational stimuli. A successful method would be important to the community as this goal has been difficult to achieve, and it would open up new avenues of research. The authors addressed some of my concerns from the last review but not all. The experiments they did perform are a screen of frequencies that damage the hair cells of the lateral line and improved the quality of an image (Figure 4). This is appreciated especially during the time of Covid.They argue that 2 suggested experiments are not necessary: microphonic potentials and damage to the core of actin of stereocilia. I feel that they would improve the paper's quality, but I see their point about not including the microphonic potentials; however, they should at some level look at the damage to the actin core. This could be by TEM as suggested or it could be by phalloidin labeling. Both of these methods can be used to see gaps in the core of actin in mouse hair cells and maybe those of zebrafish.

While we were unable to discern damage to the actin core in zebrafish neuromast hair bundles using phalloidin labeling, we were able to characterize qualitative changes as well as quantify average stereocilia length in projected 3D image stacks. We used these data to determine how stereocilia morphology corresponds with FM1-43 uptake recovery in the revised Figure 10.

Overall, the present study’s aims were to evaluate mechanical damage in the zebrafish lateral line and to establish this injury paradigm as a model for the study of synaptic repair. A more detailed analysis of actin injury and repair would require considerable TEM imaging which, while we hope may be performed in future studies, is beyond the scope of the present study.

Another issue is the use of the word "current". According to the dictionary (OED) current is defined as "That which runs or flows, a stream; spec. a portion of a body of water, or of air, etc. moving in a definite direction." This is the way I see the word used in the scientific literature as well. I don't think what the authors are providing is a current. They should use a different term, perhaps, "vibratory stimulus" or "water stimulus."

We have replaced the term “current” with “water wave” in the manuscript.